# From Per-Image Low-Rank to Encoding Mismatch: Rethinking Feature Distillation in Vision Transformers

Huiyuan Tian [1]  Bonan Xu [2]  Shijian Li [1]

## Abstract

Feature-map knowledge distillation (KD) transfers internal representations well between comparably sized Vision Transformers (ViTs), but it often fails in compression. We revisit this failure and uncover a paradox. Sample-wise SVD shows that *each image* is highly compressible, which seems to suggest that a narrow student with a linear projector should match the teacher "in principle". However, a dataset-level view contradicts this intuition: PCA shows that the teacher is a **union of low-rank subspaces** with significant subspace rotation across inputs. We further introduce token-level Spectral Energy Patterns (SEP) and find an **architecture-invariant encoding law**: tokens spread energy broadly across channel modes even when they live in low-rank subspace, creating a bandwidth mismatch. We refer to this combined phenomenon as an **encoding mismatch**. We propose two minimal remedies, **Lift** or **WideLast**: (i) **Lift** retains a lightweight lifting projector at inference to provide wider channel, or (ii) **WideLast** widens only the student's last block, enabling an input-dependent expansion. On ImageNet-1K, these fixes revive feature KD for ViT compression, improving DeiT-Tiny distilled from CaiT-S24 from 74.86% to 77.53%/78.23% top-1 accuracy, and they also strengthen students trained without distillation. Our analyses clarify when and why feature-map KD fails and then how to fix it. Code and raw data are provided in GitHub.

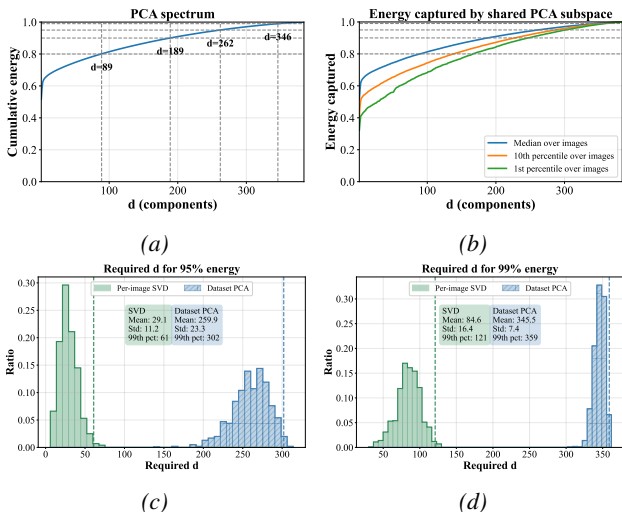

*Figure 1.* **Per-image low-rank but dataset-level non-low-rank in CaiT-S24** (Touvron et al., 2021b). (a) **Dataset-level PCA spectrum:** cumulative fraction of energy explained by the top-$d$ channel principal components learned from 1,000 images. (b) **Per-image energy captured by a shared subspace:** percentiles of $E_i(d) = \|X_i V_d\|_F^2 / \|X_i\|_F^2$ when projecting each image's token-feature matrix $X_i$ onto the same PCA basis $V_d$. (c) **Required $d$ for 95% energy:** overlay histograms comparing per-image SVD (input-dependent) vs. dataset PCA with a *shared* subspace; dashed lines mark the 99th-percentile required dimensions (61 vs. 302). (d) **Required $d$ for 99% energy:** same comparison (121 vs. 359), highlighting strong per-image compressibility but substantial subspace rotation across inputs.

## 1. Introduction

Knowledge distillation (KD) (Hinton et al., 2015) is widely used to transfer performance from large teachers to compact students. A simple and effective recipe in CNNs is *feature-map distillation*: matching intermediate representations provides strong supervision and often yields robust gains (Romero et al., 2015; Zagoruyko & Komodakis, 2017; Yim et al., 2017; Chen et al., 2021). With ViTs (Dosovitskiy et al., 2021; Caron et al., 2021; Han et al., 2022; Li et al., 2024), feature map KD is now common in two regimes. In *representation transfer*, the student has comparable capacity (e.g., CLIP-style models) (Radford et al., 2021; Feng et al., 2025), where even direct feature mimicry can work well (Yang et al., 2024a; Wu et al., 2023). In *compression*, the goal is to train a smaller student to approximate a larger teacher. In the compression setting, straightforward feature-map matching is often unreliable: it yields small

[1]Zhejiang University [2]The Hong Kong Polytechnic University. Correspondence to: Shijian Li <shijianli@zju.edu.cn>.

*Proceedings of the 43rd International Conference on Machine Learning*, Seoul, South Korea. PMLR 306, 2026. Copyright 2026 by the author(s).

gains, is sensitive to design choices, and can even hurt accuracy (Yang et al., 2024b; Miles & Mikolajczyk, 2024; Miles et al., 2024; Tian et al., 2025; 2026). Consequently, many ViT distillation methods move away from raw feature alignment toward alternative signals or mechanisms, including distillation tokens (Touvron et al., 2021a), contrastive objectives (Tian et al., 2020), attention matching (Feng et al., 2025), and architectural or pipeline modifications (Wu et al., 2022; Zhang et al., 2022; Hao et al., 2022; Fan et al., 2024). For a more comprehensive discussion of related work, see Appendix A.

What remains missing is a *simple, general, feature-map-based* distillation scheme for ViTs, together with a clear explanation of why the naive recipe fails in compression. We begin by analyzing the teacher's last-layer token feature matrix $X_i \in \mathbb{R}^{N \times D}$ for each image $i$. A sample-wise SVD view suggests strong compressibility: for CaiT-S24 (Touvron et al., 2021b) ($D = 384$, $N = 196$ patch tokens), only 61 singular directions suffice to recover 95% of the energy for $\sim 99\%$ of images (Figure 1c). This notion of "low rank" is *per-image* (as in many low-rank/redundancy observations): by the Eckart–Young–Mirsky theorem (Strang, 2022), *each* matrix $X_i$ admits an accurate rank-$d$ factorization, but its optimal subspace is generally *input-dependent*. It is therefore tempting to conclude that a narrow student plus a linear projector should match the teacher, but that conclusion implicitly assumes that a *single shared* subspace works across inputs.

To test this, we perform dataset-level PCA. Although the leading components capture substantial energy, the global PCA spectrum remains long-tailed (Figure 1a) and more importantly, Figure 1b shows that when each image is projected onto the *same* top-$d$ PCA basis, the retained energy varies substantially across inputs (median vs. lower-percentile images). The result is striking: a shared subspace must be much wider than the per-image SVD rank to preserve high teacher-feature energy. For CaiT-S24, achieving 95% energy for $\sim 99\%$ of images requires $d \approx 302/384$ (Figure 1c), and even 99% energy needs $d \approx 359/384$ (Figure 1d). Thus **individual samples are low-rank, but the dataset is not**: per-image matrices are compressible, yet their principal subspaces rotate substantially across inputs (see Appendix B for the same results across architectures). Throughout, our "low rank" refers to sample-wise low rank of token feature matrices (per image), and does not imply that the dataset shares a single low-dimensional subspace.

To understand why this mismatch persists even beyond subspace rotation, we further ask how much channel capacity is used *within* whatever subspace a token occupies. We introduce *token-level Spectral Energy Pattern* (SEP) analysis, a channel-spectrum diagnostic that measures how each token distributes energy across channel modes. SEP is basis-

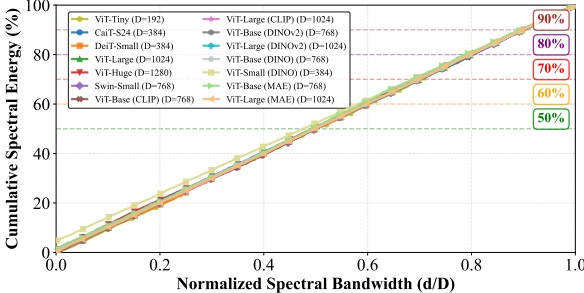

*Figure 2.* **Token-level Spectral Energy Pattern (SEP) is architecture and training regime invariant.** For each model, we take last-layer token features, apply a 1D DFT along the channel dimension, and plot the cumulative spectral energy as a function of normalized bandwidth ($d/D$), averaged over 1,000 ImageNet-1K validation images. Despite large differences in architecture and width (ViT-Tiny/DeiT-Small/CaiT-S24/ViT-Large/ViT-Huge/Swin-Small), the SEP curves nearly collapse onto a common, almost diagonal profile: capturing $\alpha \in \{50, 70, 90\}\%$ of a token's energy requires roughly $\alpha\%$ of the available channel modes. This "spectral universality" indicates high per-token channel utilization at the representation endpoint, even when per-image token matrices are highly compressible under SVD.

dependent and is not meant to assign physical frequency semantics to transformer channels. We use it as an empirical bandwidth probe and validate its qualitative behavior under random global channel permutations. SEP reveals an **architecture-invariant encoding law**: individual tokens consistently spread energy broadly across channel modes, implying a high-bandwidth per-token encoding requirement. Combined with dataset-level subspace rotation, SEP exposes an **endpoint capacity mismatch**. Each image may live in a low-dimensional subspace, but tokens still require high bandwidth inside that subspace, which a narrow student cannot reproduce. Guided by these discoveries, we propose two minimal strategies that make feature-map KD effective again for ViT compression:

1. **Lift: Post-Hoc Feature Lifting.** We attach a lightweight linear projector after the student's last layer to lift features to the teacher width, and *retain it at inference*. This fixed interface supplies near-teacher endpoint width/capacity and helps the student match the teacher's high-bandwidth token encodings.

2. **WideLast: Native Width Alignment.** We widen only the student's final Transformer block to match the teacher width. Unlike a fixed projector, a widened last block is an *input-dependent* mapping that can realize different effective orientations/subspaces across images, while also providing near-teacher endpoint width.

On ImageNet-1K, with a DeiT-Tiny student and a CaiT-S24

*Table 1.* Last-layer/stage effective dimensions (99th-percentile) across architectures. Left: per-image SVD (input-dependent). Right: dataset-level PCA with a shared subspace. SL: supervised learning; SSL: self-supervised learning; MM: multimodal pretraining.

| Training Method | Model | Dims | Per-image SVD (99th pct) | | | | Dataset PCA (shared, 99th pct) | | | |
| --- | --- | --- | --- | --- | --- | --- | --- | --- | --- | --- |
| | | | 80% | 90% | 95% | 99% | 80% | 90% | 95% | 99% |
| SL | ViT-Tiny | 192 | 1 | 2 | 7 | 52 | 1 | 17 | 71 | 154 |
| SL | CaiT-S24 | 384 | 14 | 34 | 61 | 121 | 167 | 247 | 302 | 359 |
| SL | DeiT-Small | 384 | 30 | 59 | 89 | 146 | 205 | 278 | 324 | 369 |
| SL | Swin-Small | 768 | 3 | 7 | 15 | 33 | 129 | 336 | 495 | 691 |
| SL | ViT-Large | 1024 | 5 | 16 | 39 | 115 | 255 | 498 | 694 | 933 |
| SSL | ViT-Small (DINO) | 384 | 41 | 72 | 101 | 152 | 232 | 297 | 336 | 373 |
| SSL | ViT-Base (DINO) | 768 | 39 | 75 | 109 | 165 | 382 | 529 | 624 | 731 |
| SSL | ViT-Base (DINOv2) | 768 | 19 | 39 | 67 | 146 | 384 | 519 | 606 | 704 |
| SSL | ViT-Large (DINOv2) | 1024 | 30 | 58 | 93 | 173 | 525 | 696 | 813 | 940 |
| SSL | ViT-Base (MAE) | 768 | 1 | 7 | 34 | 113 | 1 | 58 | 235 | 585 |
| SSL | ViT-Large (MAE) | 1024 | 1 | 9 | 38 | 120 | 1 | 49 | 268 | 765 |
| SSL | ViT-Huge (MAE) | 1280 | 1 | 15 | 58 | 164 | 1 | 83 | 371 | 977 |
| MM | ViT-Base (CLIP) | 768 | 6 | 27 | 60 | 135 | 129 | 340 | 502 | 704 |
| MM | ViT-Large (CLIP) | 1024 | 15 | 46 | 89 | 181 | 230 | 506 | 713 | 945 |

teacher, both strategies turn simple feature alignment from unreliable to consistently beneficial, improving Top-1 accuracy from 74.86% to 77.53% and 78.23%, respectively. Moreover, both modifications also improve the standalone student trained without a teacher, indicating that the endpoint mismatch reflects a genuine architectural bottleneck. Our main contributions are:

1. **Sample-wise vs. dataset-wise geometry at the endpoint**. We show that ViT features are compressible per image (low-rank token matrices), yet require a wide shared subspace across the dataset (dataset-level PCA), revealing substantial subspace rotation.

2. **SEP reveals high per-token bandwidth**. We introduce SEP and find an architecture-invariant high-bandwidth token encoding pattern.

3. **A revised failure mechanism for feature KD**. We connect dataset-level subspace rotation and SEP bandwidth to an endpoint capacity mismatch that explains why naive feature-map KD fails in wide-to-narrow ViT compression.

4. **Mismatch-guided minimal fixes**. We propose post-hoc lifting and last-block widening, two simple remedies that reactivate feature-map distillation and improve compact ViTs even without distillation.

## 2. Representation Analysis

We analyze the teacher's last-layer token representation from three complementary viewpoints: (1) Sample-wise

token SVD; (2) Dataset-level shared-subspace PCA; (3) Token-level SEP.

### 2.1. Sample-Wise SVD Analysis

Sample-wise token SVD quantifies per-image compressibility across tokens, which is an *input-dependent* low-rank factorization. Given a teacher feature matrix $\mathbf{X} \in \mathbb{R}^{N \times D}$ (tokens $\times$ channels) at a particular layer, its SVD is $\mathbf{X} = \mathbf{U}\mathbf{\Sigma}\mathbf{V}^\top$, where $\{\sigma_i\}$ are the singular values. Note that $\mathrm{rank}(\mathbf{X}) \leq \min(N, D)$; for ViTs with $16 \times 16$ patches, $N = 196$ (dropping [CLS]) provides rank ceiling, so we interpret "low-rank" through *effective dimension* relative to $N$. The Eckart–Young–Mirsky theorem (Strang, 2022) characterizes the optimal rank-$d$ approximation:

$$\min_{\mathbf{P}, \mathbf{Z}} \left\| \mathbf{X} - \mathbf{Z}\mathbf{P} \right\|_F^2 = \sum_{i>d} \sigma_i^2. \tag{1}$$

Here, $\mathbf{Z} \in \mathbb{R}^{N \times d}$ can be interpreted as an optimal $d$-dimensional bottleneck representation (analogous to the student's output), and $\mathbf{P} \in \mathbb{R}^{d \times D}$ as the optimal linear projector mapping that bottleneck into the teacher's space. If the tail $\sum_{i>d} \sigma_i^2$ decays rapidly, a low-rank approximation is accurate and a student equipped with a simple linear projector should be able to match the teacher's features in principle. To quantify this, we report the explained energy captured by the top $d$ singular components:

$$\mathrm{Explained}(d) = \frac{\sum_{i=1}^{d} \sigma_i^2}{\sum_{i=1}^{r} \sigma_i^2}, \tag{2}$$

where $r = \text{rank}(\mathbf{X}) \leq N$. For a target energy level $\tau$, we compute the *per-image* required dimension

$$d_i^{\text{SVD}}(\tau) = \min\{d : \text{Explained}_i(d) \geq \tau\}. \quad (3)$$

We then summarize the distribution of $d_i^{\text{SVD}}(\tau)$ over images (e.g., 50th and 99th percentiles). Importantly, Eckart–Young–Mirsky is an *existence* result for each matrix $\mathbf{X}_i$: the optimal projector corresponds to the top-$d$ right-singular vectors and is generally *input-dependent*.

## 2.2. Dataset-Level Shared-Subspace PCA

A student distillation interface (e.g., a fixed $D_S \to D_T$ linear map) corresponds to using the *same* channel subspace for all inputs. We therefore measure the dimension of such a shared subspace using **dataset-level PCA**. Given a set of last-layer feature matrices $\{\mathbf{X}_i\}_{i=1}^M$, we accumulate the channel second moment

$$\mathbf{C} = \frac{1}{T}\sum_{i=1}^M \mathbf{X}_i^\top \mathbf{X}_i, \quad T = \sum_{i=1}^M N_i. \quad (4)$$

This is equivalent to stacking all tokens across the dataset and performing PCA in channel space, but without explicitly forming the huge stacked matrix. We use the *uncentered* moment so that "energy captured" aligns with the Frobenius/SVD energy used throughout the paper. Let $\mathbf{C} = \mathbf{V}\,\text{diag}(\lambda)\,\mathbf{V}^\top$ with $\lambda_1 \geq \cdots \geq \lambda_D \geq 0$, and let $\mathbf{V}_d \in \mathbb{R}^{D \times d}$ denote the top-$d$ eigenvectors. The **global** cumulative energy explained by the shared basis is

$$E_{\text{global}}(d) = \frac{\sum_{k=1}^d \lambda_k}{\sum_{k=1}^D \lambda_k}. \quad (5)$$

More importantly for distillation, we measure how well the *same* basis $\mathbf{V}_d$ captures *each image*:

$$E_i(d) = \frac{\|\mathbf{X}_i \mathbf{V}_d\|_F^2}{\|\mathbf{X}_i\|_F^2}. \quad (6)$$

For a target energy level $\tau$, define the per-image required dimension under a shared subspace:

$$d_i^{\text{PCA}}(\tau) = \min\{d : E_i(d) \geq \tau\}. \quad (7)$$

The 99th percentile of $d_i^{\text{PCA}}(\tau)$ is the smallest width such that a *single fixed* projector $\mathbf{Q}_d = \mathbf{V}_d \mathbf{V}_d^\top$ preserves at least $\tau$ energy for $\sim 99\%$ of images.

## 2.3. Spectral Energy Pattern Analysis

The SVD and shared-PCA analyses in Sections 2.1 and 2.2 characterize subspace geometry across tokens and across inputs, but they do not reveal how much channel capacity is used **per token**. In particular, even if each image's token matrix lies near a low-dimensional subspace (sample-wise low

rank), individual tokens can still use rich, high-bandwidth encodings *within* that subspace, which is the quantity that matters for endpoint capacity. To probe per-token encoding, we introduce a token-level Spectral Energy Pattern (SEP) analysis. For each last-layer token $\mathbf{x}_t \in \mathbb{R}^D$, we compute a $D$-point Discrete Fourier Transform (DFT) along the channel dimension and preserve the full FFT order:

$$S_t(k) = \left|\text{DFT}(\mathbf{x}_t)_k\right|^2, \qquad k = 1, \ldots, D. \quad (8)$$

We then define the cumulative spectral energy pattern:

$$\text{SEP}_t(d) = 100 \cdot \frac{\sum_{k=1}^d S_t(k)}{\sum_{k=1}^D S_t(k)}, \qquad d = 1, \ldots, D, \quad (9)$$

which measures how much of the token's energy is captured by the first $d$ bins in full FFT order. To summarize how a token uses its available bandwidth, we define the normalized spectral bandwidth:

$$b_{\alpha,t} = \min\left\{\frac{d}{D} \,\middle|\, \text{SEP}_t(d) \geq \alpha\right\}. \quad (10)$$

A large $b_{\alpha,t}$ (e.g., $b_{90} \approx 0.9$) indicates that the token spreads its energy across most of the available channel modes. Because transformer channels do not have an inherent physical ordering, SEP should be interpreted as an empirical bandwidth diagnostic under a specified channel basis/order, not as a canonical frequency decomposition. SVD/PCA and SEP thus play complementary roles: SVD characterizes *per-image* compressibility, PCA characterizes the width needed for a *shared* subspace across the dataset, and SEP measures *per-token* encoding utilization within whichever subspace the image occupies. We further validate in Section C.1 that the qualitative near-diagonal SEP law is stable under random global channel permutations.

## 2.4. Low-Rank Per Image, Wide Subspace Across the Dataset

Figure 1 summarizes CaiT-S24 at the last layer. **Each image is easy to compress (sample-wise low rank).** In Figure 1c–d, per-image SVD shows that the last-layer token matrix of *each image* concentrates energy quickly: the 99th-percentile ranks are $d_{\text{SVD}}(95\%) = 61$ and $d_{\text{SVD}}(99\%) = 121$. This means that for almost all images there exists an *input-specific* low-dimensional subspace that reconstructs the teacher features well. **But the dataset is not low-rank under a shared projector (subspace rotation).** In Figure 1a–d, dataset-level PCA shows that a *single fixed* subspace must be much wider to work for most images: to preserve 95% energy for $\sim 99\%$ of images, CaiT-S24 requires $d_{\text{PCA}}(95\%) = 302/384$ dimensions (and $d_{\text{PCA}}(99\%) = 359/384$). This gap between per-image SVD and shared PCA indicates substantial *subspace rotation* across inputs.

Images live in low-dimensional subspaces, but those subspaces vary in orientation. **The pattern is universal.**

Table 1 extends this comparison across architectures and training regimes. Per-image SVD effective dimensions are often far below both $D$ and the token ceiling $N$, but dataset-level PCA requires a shared subspace that is dramatically wider and often approaches the full channel dimension. For example, ViT-Large needs only 39 dimensions per image for $95\%$ SVD energy, but needs $694/1024$ dimensions for $95\%$ energy under a shared PCA basis. Appendix B provides the corresponding four-panel visual diagnostics for these models, mirroring Figure 1. Taken together one can make a conclusion that **a single narrow fixed interface is not guaranteed to match the teacher across images**, even though each image is individually compressible.

### 2.5. Token-Level High Utilization

We now turn to SEP to assess how heavily each token uses its available channel modes. We compute the cumulative spectral energy curves at the last layer for a range of architectures and training regimes, average them over the same 1,000 ImageNet validation images used elsewhere in the paper, and plot them against the normalized bandwidth $d/D$. A large $b_\alpha$ (e.g., $b_{90} \approx 0.9$) indicates that the token spreads its energy across most of the available channel modes. Empirically, the curves in Figure 2 are remarkably similar across architectures and remain close to diagonal: approximately $50\%/60\%/70\%/80\%/90\%$ of the energy is captured by $\sim 0.50/0.60/0.70/0.80/0.90$ of the channel bandwidth, respectively. For each architecture we also report $b_\alpha$ statistics over 1,000 images and provide per-model SEP curves with mean and std bands in Section C.

Although SEP is not permutation-invariant in principle, this qualitative law is not tied to one particular native channel ordering. As a control, we apply 100 global random permutations of the channel dimension to the saved last-layer features of each model and recompute SEP. Across 14 transformer-family models, the permutation-mean SEP curves remain close to diagonal, with mean curve $L_1$ distances between 0.0012 and 0.0059; the largest absolute bandwidth shifts are $|\Delta b_{80}| = 0.0073$ and $|\Delta b_{90}| = 0.0036$. Thus the broad-band SEP conclusion is robust to arbitrary global channel reorderings; details are provided in Section C.1. This behavior has two important implications:

- **Spectral universality**. Despite differences in architecture and width $D$, last-layer tokens follow an almost universal SEP curve when expressed in normalized coordinates. This suggests a shared encoding law that tokens tend to distribute energy broadly across the available modes at the final representation.

- **High per-token utilization**. Because a large fraction of the channel spectrum is needed to capture a given fraction of energy, individual tokens are not strongly compressible under this specified channel ordering. In other words, although each image's token matrix is low-rank across tokens (Section 2.4), each token tends to use a wide set of channel modes *within whichever subspace it occupies*.

### 2.6. Two Sides of the Same Coin

Taken together, our diagnostics expose a single failure mechanism with *two coupled facets*:

- **Orientation mismatch (rotation).** With a fixed linear interface, the student must map into a channel subspace that varies across inputs, which is ill-conditioned when the student is much narrower than the teacher.

- **Capacity mismatch (bandwidth).** Even when the teacher is low-rank for a given input, the student needs near-teacher channel bandwidth to reproduce the teacher's token-level encoding pattern.

We refer to this combined phenomenon as an **encoding mismatch at the distillation endpoint**. It clarifies the new paradox: *each image is easy to compress, yet a narrow student with a fixed linear interface cannot reliably match the teacher across images*. It also suggests why our two strategies behave differently: **Lift** mainly addresses endpoint bandwidth by providing a fixed lifted interface, while **WideLast** provides both near-teacher bandwidth *and* an input-dependent mapping that can realize different effective subspace orientations.

**Connection to CNNs.** The same endpoint diagnostics can be applied to CNN final-stage feature maps by treating spatial locations as the token axis and channels as the feature axis. As summarized in Appendix E, ResNet-18/34/50/101 show the same qualitative endpoint signature under this spatial-location $\times$ channel analysis. The architectural implication, however, is different: many standard CNN distillation pairs are already endpoint-aligned (e.g., ResNet-18/34 at 512 final channels and ResNet-50/101 at 2048), which helps explain why final-layer feature matching can be more reliable in common CNN settings. A small FitNet control in Table 11 is consistent with this view, although the matched-vs.-mismatched gap is modest. The detailed CNN figures, table, and discussion remain in Appendix E as supporting context rather than a primary claim.

## 3. Method: Mismatch-Guided Feature KD

Based on these insights, we propose two minimal, interpretable strategies to resolve this encoding mismatch and **reactivate simple feature-map distillation for ViTs**.

### 3.1. Lift: Post-Hoc Feature Lifting

**Goal.** Lift targets the *endpoint bandwidth/capacity* side of the mismatch. We attach a lightweight linear *lifting* module after the student's last layer to produce a near-teacher-width representation at the distillation interface, and we *retain it at inference*. This gives the student extra endpoint capacity and lets feature losses act in the teacher coordinate system. The lift is *fixed* across inputs (so it does not resolve dataset-level subspace rotation), but empirically it is already sufficient to make simple feature-map KD reliably beneficial.

**Architecture**. Let the student's last layer output $\mathbf{X}_S \in \mathbb{R}^{N \times D_S}$, where $N$ is the number of tokens and $D_S < D_T$ is the student width. We attach a token-wise linear projector $\mathbf{P} \in \mathbb{R}^{D_S \times D_T}$ to the student's final layer and obtain the lifted features:

$$\widehat{\mathbf{X}}_S = \mathbf{X}_S \mathbf{P}. \tag{11}$$

This projector increases the student's channel capacity at the last layer while remaining linear and parameter-efficient.

**Inference**. Critically, the projector $\mathbf{P}$ is **retained at inference**. The student's classification head $\mathbf{W}_{\text{head}} \in \mathbb{R}^{D_T \times C}$ (where $C$ is the number of classes) operates on the lifted representation. We consider two common choices: applying Global Average Pooling (GAP) to the lifted token features $\widehat{\mathbf{X}}_S$, or using the lifted [CLS] token (also mapped by $\mathbf{P}$) as input to the classifier.

### 3.2. WideLast: Native Width Alignment

While post-hoc lifting addresses endpoint bandwidth with a fixed interface, our analysis in Section 2 suggests a stronger native remedy: provide near-teacher width *inside* the student at the last block. A widened last block is an *input-dependent* mapping, so it can also realize different effective orientations/subspaces across inputs, unlike a single fixed projector.

**Architecture**. We modify the student by replacing only its final Transformer block with one that matches the teacher's width $D_T$, while keeping all earlier blocks at width $D_S$. This yields a student whose last block outputs $\widetilde{\mathbf{X}}_S \in \mathbb{R}^{N \times D_T}$. Intuitively, the preceding narrow blocks compress and process features, and the final widened block re-expands them into a representation with sufficient channel capacity. Because this last block contains attention and an MLP at width $D_T$, it can implement an *input-dependent* expansion that behaves like a data-dependent projector, adapting to the rotating subspaces revealed by dataset-level PCA.

**Inference**. The classification head $\mathbf{W}_{\text{head}} \in \mathbb{R}^{D_T \times C}$ is applied directly to the output of this new, wider final block. Overall, WideLast builds the **required endpoint bandwidth** and **input-dependent channel orientation** directly into the architecture, making the student intrinsically better

*Table 2.* Main distillation results on ImageNet-1K (teacher: CaiT-S24; student: DeiT-Tiny). **Lift** (post-hoc lifting projector retained at inference) and **WideLast** (widening only the last Transformer block). SoftKD denotes logit distillation and SpecKD denotes SpectralKD.

| Variant | Distill. | Top-1 (%) | $\Delta$ |
|---|---|---|---|
| Baseline | None | 74.86 | – |
|  | SpecKD | 75.07 | +0.21 |
| Lift | SoftKD | 77.23 | +2.37 |
|  | MSE | 76.61 | +1.75 |
|  | SpecKD | 76.40 | +1.54 |
|  | SoftKD+MSE | 77.50 | +2.64 |
|  | SoftKD+SpecKD | **77.53** | **+2.67** |
| WideLast | SoftKD | 77.88 | +3.02 |
|  | MSE | 77.15 | +2.29 |
|  | SpecKD | 76.53 | +1.67 |
|  | SoftKD+SpecKD | 78.16 | +3.30 |
|  | SoftKD+MSE | **78.23** | **+3.37** |

matched to the teacher at the distillation endpoint.

### 3.3. Overall Objective

Both strategies share the same training objective. The total loss $\mathcal{L}$ is a weighted sum of the standard Cross-Entropy (CE) loss, an optional logit-based KD loss, and feature distillation loss $\mathcal{L}_{\text{feat}}$:

$$\begin{aligned} \mathcal{L} = &\, (1 - \lambda_{\text{logit}}) \, \mathcal{L}_{\text{CE}} \left( \mathbf{y}, \mathbf{p}_S \right) \\ &+ \lambda_{\text{logit}} \, \mathcal{L}_{\text{KD}} \left( \mathbf{p}_S, \mathbf{p}_T; \tau \right) + \lambda_{\text{feat}} \, \mathcal{L}_{\text{feat}} , \end{aligned} \tag{12}$$

where $\mathbf{p}_S$ and $\mathbf{p}_T$ are the student and teacher logits, $\mathbf{y}$ is the ground-truth label, and $\tau$ is the distillation temperature. The feature loss $\mathcal{L}_{\text{feat}}$ can be MSE loss or other variants like SpectralKD (Tian et al., 2025) to demonstrate generality.

## 4. Experiments

We empirically validate our analysis and mismatch-guided strategies on ImageNet-1K. We further test robustness to common design choices (classifier head, projector width, and feature-loss weight), examine whether the same effects hold across different teacher architectures, evaluate whether the proposed architectural changes improve students even without any teacher, and report parameter/compute overheads with a parameter-matched control in Appendix D.

### 4.1. Experimental Setup

**Dataset**. We conduct all experiments on ImageNet-1K (Deng et al., 2009), which contains 1.28M training images and 50K validation images across 1,000 classes. All images are resized to $224 \times 224$ following standard practice

*Table 3.* Ablation on head type. Gains are measured relative to each head's own standalone baseline.

| Head Type | KD Method | Top-1 (%) | Δ |
|-----------|-----------|-----------|-----|
| [CLS] Token | None | 74.60 | – |
| | MSE Only | 75.43 | +0.83 |
| | SpectralKD Only | 75.72 | +1.12 |
| | SoftKD (Logits) | 76.62 | +2.02 |
| GAP | None | 75.41 | – |
| | MSE Only | 76.61 | +1.20 |
| | SpectralKD Only | 76.40 | +0.99 |
| | SoftKD (Logits) | 77.23 | +1.82 |

*Table 4.* Ablation on projector output dimension $D_T$. Baseline (192-dim) scores: 74.86% standalone, 75.07% with SpectralKD.

| Projector Dims | w/o KD | SpectralKD Only |
|----------------|--------|-----------------|
| 256 | 75.46 | 76.10 |
| 320 | 75.53 | 76.24 |
| 384 (Teacher) | 75.41 | 76.40 |
| 448 | 75.23 | 76.48 |

(Dosovitskiy et al., 2021; Touvron et al., 2021a).

**Models**. We adopt DeiT-Tiny (Touvron et al., 2021a) as the student baseline (5.7M parameters, 192 channels, 12 layers). The primary teacher is CaiT-S24 (Touvron et al., 2021b) (47M parameters, 384 channels, 24 self-attention + 2 class-attention layers). We also evaluate generality with DeiT-Small and DeiT3-Small (both 384 channels) in Section 4.4. All pretrained teachers are loaded from the timm library (Wightman, 2019); students are trained from scratch.

**Training protocol**. We follow the DeiT training recipe (Touvron et al., 2021a). All models are trained for 300 epochs with AdamW (Loshchilov & Hutter, 2019), base learning rate $5 \times 10^{-4}$, weight decay 0.05, cosine decay schedule (Loshchilov & Hutter, 2017), 5-epoch linear warmup from $1 \times 10^{-6}$. The effective batch size is 2048 (512 per GPU on 4×NVIDIA RTX 4090 GPUs). We use PyTorch distributed data parallel (Paszke et al., 2019).

### 4.2. Main Results: Reactivating Feature-Map KD

We first distill a CaiT-S24 teacher (384-dim) into a DeiT-Tiny student (192-dim). Results are summarized in Table 2.

**Baseline and failure of naive feature KD**. The standard DeiT-Tiny baseline trained with cross-entropy reaches 74.86% Top-1 accuracy. Applying a simple feature KD method (SpectralKD (Tian et al., 2025)) directly to this baseline only improves Top-1 to 75.07% (+0.21). This confirms the empirical phenomenon reported in prior work (Yang et al., 2024b; Tian et al., 2026) that plain feature-map

*Table 5.* Ablation on feature loss weight $\lambda_{\text{feat}}$.

| KD Method | $\lambda_{\text{feat}}$ | Top-1 (%) |
|-----------|--------|-----------|
| SpectralKD Only | 0.2 | 76.40 |
| SpectralKD Only | 0.3 | 76.74 |
| MSE Only | 0.2 | 76.61 |
| MSE Only | 0.3 | 76.72 |
| SoftKD + MSE | 0.2 | 77.15 |
| SoftKD + MSE | 0.3 | 77.50 |

distillation is almost ineffective.

**Lift (post-hoc lifting projector)**. We attach a lightweight linear projector ($192 \rightarrow 384$) after the student's last layer and keep it at inference. Even without feature KD, this modification plus SoftKD on logits already achieves 77.23%. Crucially, the same projector **reactivates simple feature-map alignment**: (1) MSE-only feature KD (which fails on the original student) now yields 76.61% (+1.75 over baseline); (2) SpectralKD reaches 76.40%; (3) Combining SoftKD with MSE or SpectralKD further improves performance to 77.50% and 77.53%, respectively. Thus, once the encoding mismatch is removed, even a vanilla MSE feature loss becomes reliably beneficial.

**WideLast (native width alignment)**. We then explicitly match the student's final-layer width to the teacher by replacing only the last Transformer block with a 384-dim block (all earlier layers remain 192-dim). This yields even stronger feature KD: (1) MSE-only feature alignment now reaches 77.15% (+2.29); (2) SoftKD alone achieves 77.88%; (3) Combining SoftKD with SpectralKD or MSE pushes performance to 78.16% and 78.23%. These results directly validate our analysis: **once the student is given sufficient token-level encoding capacity, simple feature-map MSE becomes a strong distillation signal**, closing the "low-rank yet hard to distill" paradox observed in Section 2.

### 4.3. Ablation Studies

We next analyze how our design behaves under different classifier heads, projector widths, and feature loss weights, to test robustness and better understand the role of mismatch.

#### 4.3.1. VERSATILITY ACROSS CLASSIFICATION HEADS

For Lift, the lifted features can feed the classifier using either the [CLS] token or GAP over patch tokens (Table 3). This ablation tests whether our gains are tied to a specific head design. In both cases, our feature KD becomes effective once the projector resolves the encoding mismatch. This shows that our mismatch-guided strategy is agnostic to the choice of common classifier heads. Since GAP is slightly stronger here, we adopt it in all other experiments of Lift.

*Table 6.* Distillation from an ImageNet-1K DeiT-Small teacher to DeiT-Tiny. The direct feature-KD row uses the plain DeiT-Tiny student without **Lift** or **WideLast**.

| Model Configuration | Teacher | Feature KD | Top-1 (%) | Δ vs. Baseline |
|---|---|---|---|---|
| DeiT-Tiny (Baseline) | None | None | 74.86 | – |
| Plain DeiT-Tiny | DeiT-Small | SpectralKD Only | 74.52 | -0.34 |
| Lift | DeiT-Small | MSE Only | 76.22 | +1.36 |
| Lift | DeiT-Small | SpectralKD Only | 75.64 | +0.78 |
| WideLast | DeiT-Small | MSE Only | 75.66 | +0.80 |
| WideLast | DeiT-Small | SpectralKD Only | 75.73 | +0.87 |

*Table 7.* Distillation from an ImageNet-21K DeiT3-Small teacher to DeiT-Tiny. The feature-loss weight $\lambda_{\text{feat}}$ is shown because this teacher exhibits stronger sensitivity to the weight scale.

| Model Configuration | Teacher | Feature KD | $\lambda_{\text{feat}}$ | Top-1 (%) | Δ vs. Baseline |
|---|---|---|---|---|---|
| DeiT-Tiny (Baseline) | None | None | – | 74.86 | – |
| Plain DeiT-Tiny | DeiT3-Small 21k | SpectralKD Only | 0.001 | 74.10 | -0.76 |
| Plain DeiT-Tiny | DeiT3-Small 21k | SpectralKD Only | 0.005 | 72.80 | -2.06 |
| Plain DeiT-Tiny | DeiT3-Small 21k | SpectralKD Only | 0.2 | 71.72 | -3.14 |
| Lift | DeiT3-Small 21k | MSE Only | 0.001 | 75.64 | +0.78 |
| Lift | DeiT3-Small 21k | SpectralKD Only | 0.001 | 75.52 | +0.66 |
| Lift | DeiT3-Small 21k | MSE Only | 0.005 | 75.88 | +1.02 |
| Lift | DeiT3-Small 21k | SpectralKD Only | 0.005 | 75.86 | +1.00 |
| Lift | DeiT3-Small 21k | MSE Only | 0.2 | 74.77 | -0.09 |
| Lift | DeiT3-Small 21k | SpectralKD Only | 0.2 | 73.84 | -1.02 |
| WideLast | DeiT3-Small 21k | MSE Only | 0.001 | 74.97 | +0.11 |
| WideLast | DeiT3-Small 21k | SpectralKD Only | 0.001 | 75.42 | +0.56 |
| WideLast | DeiT3-Small 21k | MSE Only | 0.005 | 76.59 | +1.73 |
| WideLast | DeiT3-Small 21k | SpectralKD Only | 0.005 | 76.06 | +1.20 |
| WideLast | DeiT3-Small 21k | MSE Only | 0.2 | 75.82 | +0.96 |
| WideLast | DeiT3-Small 21k | SpectralKD Only | 0.2 | 76.50 | +1.64 |

### 4.3.2. PROJECTOR OUTPUT DIMENSION

Our analysis suggests that the student should be lifted into the teacher's subspace rather than arbitrarily wider spaces. To test this, we vary the projector output dimension $D_T \in \{256, 320, 384, 448\}$ and evaluate both standalone performance and SpectralKD performance (Table 4). All projector sizes improve over the 192-dim baseline (74.86%), peaking at 320-dim (75.53%) and the teacher-matched 384-dim (75.41%). Pushing beyond the teacher width to 448-dim actually hurts standalone performance (75.23%). While 448-dim obtains a slightly higher SpectralKD result (76.48%), it does so from a weaker standalone model and does not support simple MSE KD as robustly. These observations support our interpretation: the goal is to match the teacher's subspace, not to blindly increase dimensionality.

### 4.3.3. FEATURE LOSS WEIGHT

Finally, we study the sensitivity to the feature loss weight $\lambda_{\text{feat}}$ in the total objective (Table 5). Across both MSE-only and SpectralKD-only settings, increasing $\lambda_{\text{feat}}$ from 0.2 to 0.3 consistently yields moderate but stable improvements. Our main claims do not rely on aggressive tuning, feature KD is consistently useful once the encoding mismatch is resolved.

### 4.4. Generality Across Teachers

To test whether our conclusions depend on the teacher architecture, we repeat the core experiments using two 384-dimensional teachers: DeiT-Small trained on ImageNet-1K and DeiT3-Small pretrained on ImageNet-21K. We report them separately because the ImageNet-21K teacher intro-

*Table 8.* Standalone ImageNet-1K performance (no distillation) of DeiT-Tiny variants. **Lift** uses a post-hoc lifting projector and **WideLast** widens only the last Transformer block.

| Variant | Top-1 (%) | $\Delta$ |
|---------|-----------|----------|
| Baseline | 74.86 | – |
| Lift | 75.41 | +0.55 |
| WideLast | 75.54 | +0.68 |

duces a dataset/pretraining mismatch and is substantially more sensitive to the feature-loss weight.

With the ImageNet-1K DeiT-Small teacher (Table 6), direct feature-map distillation on the plain DeiT-Tiny student is not a strong baseline: SpectralKD-only reaches 74.52%, below the 74.86% baseline. Once the endpoint mismatch is addressed, feature KD becomes consistently useful. Lift achieves 76.22% with MSE-only KD and 75.64% with SpectralKD-only, while WideLast reaches 75.66% and 75.73%, respectively. These results show that the benefit is not specific to CaiT-S24.

The DeiT3-Small-21k results (Table 7) are qualitatively different. Directly distilling the plain DeiT-Tiny student from this ImageNet-21K teacher gives negative transfer across the tested weights (74.10%, 72.80%, and 71.72% for $\lambda_{\text{feat}} = 0.001, 0.005, 0.2$). Lift and WideLast still recover positive results at small or moderate weights, but the large weight 0.2 becomes unstable for several configurations. For example, in the Lift + MSE-only setting with $\lambda_{\text{feat}} = 0.2$, the feature loss starts at 460.6 at epoch 0, remains 311.1 at epoch 10, and is still 56.2 at epoch 299. This scale mismatch suggests that ImageNet-1K may occupy only part of the endpoint encoding space learned by the 21K teacher, so strong feature-map alignment can over-constrain a student whose final task is ImageNet-1K classification. We therefore treat dataset/pretraining mismatch in feature-level KD as an open issue, while the positive Lift/WideLast rows still support the endpoint-mismatch diagnosis.

### 4.5. Standalone Performance Without Distillation

Our analysis in Section 2 predicts that the standard DeiT-Tiny architecture is bottlenecked at its last layer due to insufficient token-level encoding capacity. If true, then **fixing this mismatch should improve the student even without any teacher or distillation signal**.

Table 8 confirms this prediction. When trained from scratch without KD, the original DeiT-Tiny reaches 74.86%, while adding our post-hoc projector (Lift) improves accuracy to 75.41% (+0.55), and widening only the final block to match the teacher width (WideLast) further raises performance to 75.54% (+0.68). These consistent gains, achieved without using any teacher, show that the encoding mismatch is a **true**

architectural bottleneck rather than an artifact of distillation. Resolving it both (i) makes feature-map KD effective again and (ii) yields stronger compact ViTs in their standalone form, *offering a simple and interpretable design cue for future compact architectures*. Parameter/FLOP details and the Uniform219dim control are reported in Appendix D.

## 5. Conclusion

We investigate why straightforward feature-map distillation, despite its success in CNNs, often fails when compressing a wide Vision Transformer into a narrower one. Sample-wise SVD shows that *each image* is highly compressible (low effective rank across tokens), yet dataset-level PCA shows that a *single shared* channel subspace must be almost full width for high-fidelity reconstruction, implying substantial *subspace rotation* across inputs. SEP further reveals that, within whichever subspace a token occupies, it uses high bandwidth across channel modes. Together these results explain why a narrow student with a fixed linear interface cannot reliably match teacher features across images: the interface is both *too narrow* (insufficient endpoint bandwidth) and *too rigid* (cannot adapt to rotating subspace orientations).

Guided by this diagnosis, we propose two minimal remedies that revitalize simple feature alignment: (i) **Lift**, a lightweight post-hoc lifting projector retained at inference, and (ii) **WideLast**, native width alignment by widening only the student's last block. On ImageNet-1K, both strategies consistently reactivate feature-map distillation (even with a vanilla MSE loss), improving DeiT-Tiny distilled from CaiT-S24 from 74.86% to 77.53%/78.23% top-1 accuracy, while also strengthening the student when trained without a teacher. These findings provide a practical design cue for compact ViTs: when distillation targets the representation endpoint, success depends on both *endpoint capacity* and the ability to realize *input-dependent* channel orientations.

Several directions remain open. First, while we focus on the last-layer mismatch that dominates wide to narrow compression, applying this dual-view analysis to intermediate layers may clarify when earlier-layer feature KD becomes beneficial. Second, adaptive or input-dependent lifting (or selectively widened blocks) may offer a superior accuracy–efficiency trade-off compared to a fixed projector. Finally, it is valuable to investigate whether analogous encoding mismatches persist in other transformer settings (e.g., detection/segmentation backbones, multimodal encoders), and whether similarly minimal capacity fixes systematically improve representation-level distillation in those domains.

## Acknowledgments

This work was supported by the Zhejiang Provincial Natural Science Foundation of China (grant LD24F030002).

## Impact Statement

This paper presents work whose goal is to advance the field of Machine Learning. There are many potential societal consequences of our work, none which we feel must be specifically highlighted here.

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

# A. Related Work

## A.1. Feature-Map KD Is Brittle in Wide-to-Narrow ViT Compression

Beyond logit distillation (Hinton et al., 2015), transferring intermediate representations has long been effective in CNNs (Romero et al., 2015; Zagoruyko & Komodakis, 2017; Yim et al., 2017; Chen et al., 2021). In ViT *compression*, however, directly matching hidden states (especially at the endpoint) is frequently unstable: it can yield only marginal gains or even degrade accuracy when the teacher is substantially wider or architecturally heterogeneous (Yang et al., 2024b; Miles et al., 2024; Miles & Mikolajczyk, 2024; Tian et al., 2025; 2026). As a result, many recent ViT distillation pipelines either (i) add dedicated alignment capacity, or (ii) distill alternative signals (tokens/attention/spectra) instead of relying on a single raw feature match (Touvron et al., 2021a; Zhang et al., 2022; Hao et al., 2022; Fan et al., 2024; Feng et al., 2025). Our work targets this common failure case and provides a concrete mechanism: a narrow student can fail at the distillation interface because it must match a teacher endpoint that is simultaneously *rotating across images* (dataset-level subspace rotation) and *high-bandwidth per token* (SEP).

## A.2. Representation Geometry, Redundancy, and the "Low-Rank but Hard to Distill" Paradox

A broad literature observes redundancy in modern networks, motivating pruning, low-rank factorization, and efficient architectures. GhostNet (Han et al., 2020), for example, argues that many feature-map channels can be generated from a smaller set of "intrinsic" maps via cheap transformations, highlighting that raw channel counts can overstate the effective degrees of freedom. Yu and Wu (Yu & Wu, 2023) similarly report that Transformer *features* can be strongly low-rank even when the corresponding *weights* are not, and leverage this observation for few-shot low-rank compression. However, redundancy alone does not explain why *feature matching can still fail* even when representations appear compressible. Our SVD+PCA+SEP analysis resolves this paradox by separating notions that are conflated in purely global redundancy arguments: (i) each image's token matrix can be low-rank, yet (ii) a *shared* subspace that works for most images can still be nearly full-width (subspace rotation), while (iii) individual tokens can still use most available channel modes within their subspace (high SEP bandwidth). This combination directly predicts when wide-to-narrow endpoint distillation is ill-conditioned.

## A.3. Bridging the Distillation Interface via Explicit Alignment

A prominent line of successful ViT feature distillation explicitly builds a stronger compatibility map between teacher and student rather than assuming their hidden states are directly comparable. ScaleKD (Fan et al., 2024) is representative: it targets cross-architecture/scale settings and introduces scale-aware alignment components (e.g., learned projectors and structured feature mimicry) so supervision is applied in a space where teacher information is accessible despite mismatched widths and inductive biases. Related methods similarly stabilize feature transfer via explicit projections/normalization (e.g., VkD (Miles et al., 2024)) or by projecting heterogeneous intermediate features into an aligned latent space (e.g., OFA-KD (Hao et al., 2023)). Our two strategies are complementary but deliberately minimal: instead of building a large translator module, we *repair the endpoint capacity/width bottleneck* (post-hoc lifting retained at inference, or last-block widening), which is sufficient to make even vanilla MSE feature matching reliable in the wide-to-narrow regime we study.

## A.4. Choosing What and Where to Distill: Tokens, Attention, and Spectral Views

Another class of approaches improves robustness by changing *which* layers/features are distilled and how they are compared. ViTKD (Yang et al., 2024b) reports depth-dependent behavior and motivates selective layer strategies, while SpectralKD (Tian et al., 2025) argues that spectral views can guide both layer selection and alignment. Attention-based objectives and hybrid schemes (e.g., MiniViT (Zhang et al., 2022)) also combine hidden-state and attention/token supervision rather than relying on a single last-layer match. These methods are compatible with our findings: they implicitly avoid concentrating all supervision on a single, potentially mismatched endpoint, and our analysis clarifies *why* the endpoint is problematic when the student is much narrower.

## A.5. Conditioning the Teacher Signal via Redundancy Suppression

Redundancy Suppression Distillation (RSD) (Zhang et al., 2025) improves cross-architecture feature transfer by conditioning the teacher signal itself, combining invariance-maximization and feature-decorrelation style objectives to extract more architecture-agnostic knowledge. Mechanistically, such teacher-conditioning methods can be viewed as making the

transferred supervision lower-redundancy and thus easier for a constrained student to absorb. This perspective aligns with our diagnosis: if the endpoint target is effectively high-bandwidth, either the interface must be strengthened (alignment modules or our endpoint fixes) or the teacher signal must be conditioned/compressed to match the student's degrees of freedom.

## B. Cross-Model SVD/PCA Diagnostics

Table 1 summarizes last-layer (or last-stage) effective dimensions for several transformer families using two complementary notions: *per-image* SVD (an input-dependent best subspace) and *dataset-level* PCA (a single shared subspace across images). Figure 1 visualizes this comparison for CaiT-S24. Here we repeat the same four-panel diagnostic for additional models spanning supervised learning (SL), self-supervised learning (SSL), and multimodal pretraining (MM), to make the phenomenon concrete. For each model, we show: (a) the dataset-level PCA spectrum; (b) percentiles of per-image energy captured by the top-$d$ *shared* PCA subspace; (c) overlay histograms of the required dimension $d$ to reach $95\%$ energy under per-image SVD vs. shared PCA; and (d) the analogous comparison for $99\%$ energy.

Across all models in Figures 3 to 15, we observe the same signature: **each image is individually low-rank**, yet a **single fixed subspace** that works for almost all images must be dramatically wider, often approaching the full channel dimension. This consistent gap supports the subspace-rotation view in Section 2.4 and indicates that the endpoint *encoding mismatch* is not specific to CaiT, but a broad property of modern ViT representations.

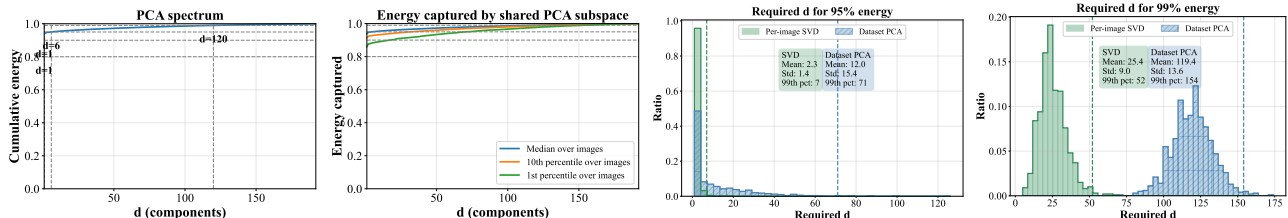

*Figure 3.* ViT-Tiny.

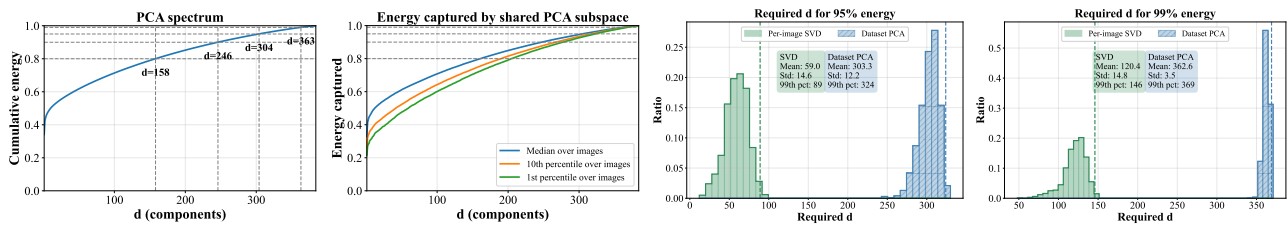

*Figure 4.* DeiT-Small.

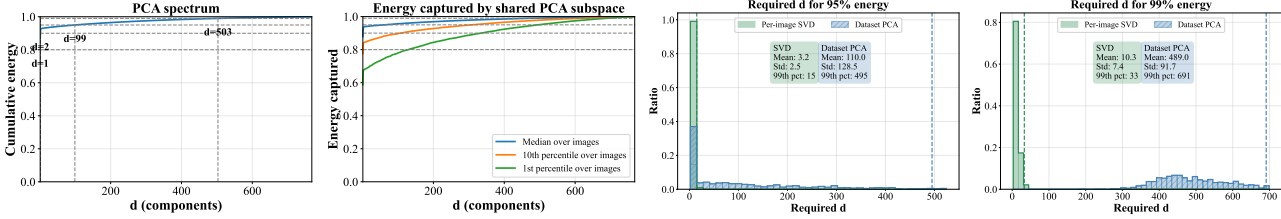

*Figure 5.* Swin-Small.

## C. SEP Statistics and Per-Model Curves

Section 2.5 introduces the token-level Spectral Energy Pattern (SEP) and shows that different architectures share a nearly identical SEP curve when plotted in normalized bandwidth coordinates. Figure 2 overlays mean SEP curves across models but omits variability. To make the underlying statistics explicit, Figure 16 and Figure 17 plot, for each architecture separately,

the cumulative spectral energy as a function of normalized bandwidth $d/D$ together with a ±1 standard-deviation band over 1,000 ImageNet-1K validation images.

Across ViT-Tiny, CaiT-S24, DeiT-Small, Swin-Small, ViT-Large, ViT-Small, ViT-Small (DINO), ViT-Base (DINO), ViT-Base (DINOv2), ViT-Large (DINOv2), ViT-Base (MAE), ViT-Large (MAE), ViT-Huge (MAE), ViT-Base (CLIP), and ViT-Large (CLIP), the SEP curves remain almost perfectly diagonal: capturing 50%, 60%, 70%, 80%, or 90% of a token's energy consistently requires ≈ 50%, 60%, 70%, 80%, or 90% of the available channel modes. The shaded bands are very narrow, indicating that this behavior is highly stable across images.

For each model, we compute the normalized bandwidth $b_\alpha$ according to Equation (10). For example, at target energy levels $\alpha = 80\%$ and $\alpha = 90\%$, the resulting means cluster tightly around $b_{80} \approx 0.8$ and $b_{90} \approx 0.9$ for all architectures. This confirms both spectral universality (similar SEP across backbones) and high per-token utilization (tokens spread their energy broadly over channel modes), even though per-image token matrices are highly compressible under SVD (and shared-PCA reveals substantial subspace rotation). Table 9 reports the corresponding normalized bandwidths $b_\alpha$ (and the absolute effective dimensions $d_\alpha = b_\alpha D$) for all models shown in Figure 16 and Figure 17.

These detailed SEP statistics support our central claim: although per-image token matrices are low-rank (and their subspaces rotate across inputs), each token still consumes most of the available channel bandwidth, so a narrow student lacks the capacity to reproduce the teacher's high-bandwidth encoding without either **Lift** or **WideLast** at the last layer.

*Table 9.* Normalized spectral bandwidth $b_\alpha$ and effective dimension $d_\alpha$ at $\alpha \in \{50, \ldots, 90\}$ for the last-layer SEP of different architectures. $D$ denotes the channel dimension.

| Model | $D$ | $b_{50}$ | $d_{50}$ | $b_{60}$ | $d_{60}$ | $b_{70}$ | $d_{70}$ | $b_{80}$ | $d_{80}$ | $b_{90}$ | $d_{90}$ |
|---|---|---|---|---|---|---|---|---|---|---|---|
| ViT-Tiny | 192 | 0.505 | 97 | 0.604 | 116 | 0.703 | 135 | 0.797 | 153 | 0.901 | 173 |
| CaiT-S24 | 384 | 0.500 | 192 | 0.599 | 230 | 0.703 | 270 | 0.805 | 309 | 0.901 | 346 |
| DeiT-Small | 384 | 0.503 | 193 | 0.602 | 231 | 0.701 | 269 | 0.797 | 306 | 0.901 | 346 |
| Swin-Small | 768 | 0.501 | 385 | 0.599 | 460 | 0.701 | 538 | 0.799 | 614 | 0.901 | 692 |
| ViT-Large | 1024 | 0.500 | 512 | 0.599 | 613 | 0.700 | 717 | 0.801 | 820 | 0.899 | 921 |
| ViT-Small (DINO) | 384 | 0.477 | 183 | 0.583 | 224 | 0.690 | 265 | 0.792 | 304 | 0.896 | 344 |
| ViT-Base (DINO) | 768 | 0.496 | 381 | 0.598 | 459 | 0.701 | 538 | 0.798 | 613 | 0.901 | 692 |
| ViT-Base (DINOv2) | 768 | 0.500 | 384 | 0.600 | 461 | 0.702 | 539 | 0.802 | 616 | 0.900 | 691 |
| ViT-Large (DINOv2) | 1024 | 0.497 | 509 | 0.600 | 614 | 0.699 | 716 | 0.799 | 818 | 0.899 | 921 |
| ViT-Base (MAE) | 768 | 0.493 | 379 | 0.589 | 452 | 0.685 | 526 | 0.789 | 606 | 0.896 | 688 |
| ViT-Large (MAE) | 1024 | 0.500 | 512 | 0.597 | 611 | 0.697 | 714 | 0.798 | 817 | 0.900 | 922 |
| ViT-Huge (MAE) | 1280 | 0.494 | 632 | 0.593 | 759 | 0.695 | 889 | 0.800 | 1024 | 0.901 | 1153 |
| ViT-Base (CLIP) | 768 | 0.500 | 384 | 0.596 | 458 | 0.698 | 536 | 0.805 | 618 | 0.901 | 692 |
| ViT-Large (CLIP) | 1024 | 0.499 | 511 | 0.602 | 616 | 0.696 | 713 | 0.802 | 821 | 0.900 | 922 |

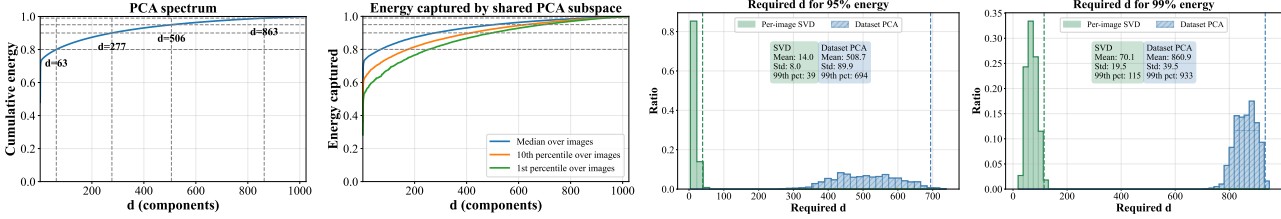

*Figure 6.* ViT-Large.

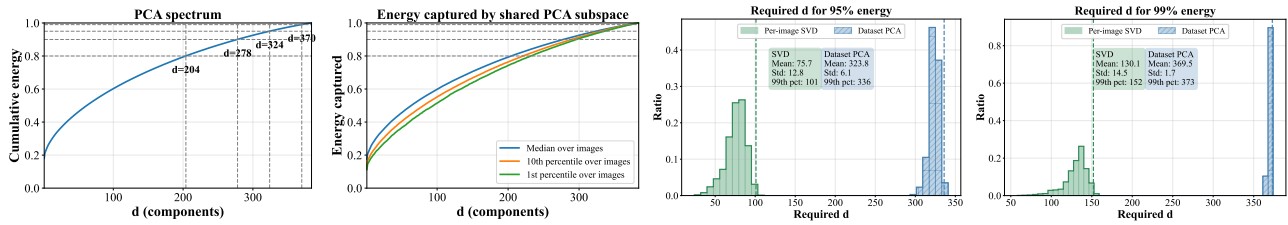

*Figure 7.* ViT-Small (DINO).

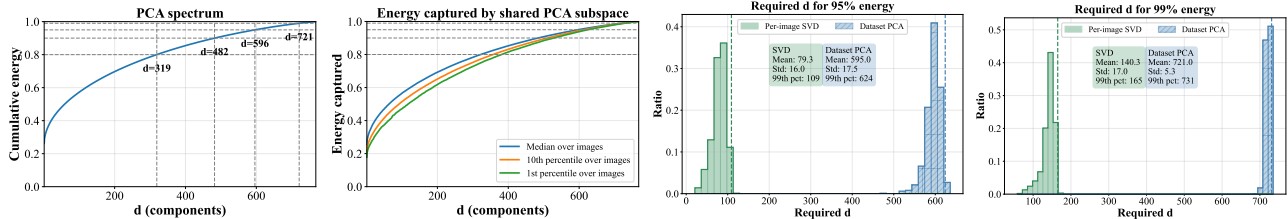

*Figure 8.* ViT-Base (DINO).

## C.1. Permutation Robustness of SEP

SEP is basis-dependent and is not mathematically invariant to arbitrary channel permutations. To test whether the near-diagonal SEP law depends on the native channel ordering, we perform a post-hoc control on the saved last-layer features. For each model, we sample $R = 100$ global random permutations $\pi_r$ of the $D$ channel indices and apply the same permutation to every last-layer token of every image in the 1,000-image validation set. No retraining is involved.

Let image $i$ contain last-layer tokens $\mathbf{x}_{i,t} \in \mathbb{R}^D$, with $t = 1, \ldots, N_i$. For permutation trial $r$, define

$$\mathbf{x}_{i,t}^{(\pi_r)} = \mathbf{\Pi}_{\pi_r} \mathbf{x}_{i,t}, \tag{13}$$

where $\mathbf{\Pi}_{\pi_r}$ is the permutation matrix corresponding to $\pi_r$. We then compute

$$S_{i,t}^{(\pi_r)}(k) = \left| \mathrm{FFT}(\mathbf{x}_{i,t}^{(\pi_r)})_k \right|^2, \qquad k = 1, \ldots, D, \tag{14}$$

$$\mathrm{SEP}_{i,t}^{(\pi_r)}(d) = 100 \cdot \frac{\sum_{k=1}^{d} S_{i,t}^{(\pi_r)}(k)}{\sum_{k=1}^{D} S_{i,t}^{(\pi_r)}(k)}, \tag{15}$$

and the image-level SEP curve

$$\mathrm{SEP}_i^{(\pi_r)}(d) = \frac{1}{N_i} \sum_{t=1}^{N_i} \mathrm{SEP}_{i,t}^{(\pi_r)}(d). \tag{16}$$

The corresponding normalized bandwidth is

$$b_{\alpha,i}^{(\pi_r)} = \min \left\{ \frac{d}{D} \,\middle|\, \mathrm{SEP}_i^{(\pi_r)}(d) \geq \alpha \right\}. \tag{17}$$

We compare the original mean SEP curve

$$\bar{\mathrm{SEP}}(d) = \frac{1}{M} \sum_{i=1}^{M} \mathrm{SEP}_i(d) \tag{18}$$

to the permutation mean curves

$$\bar{\mathrm{SEP}}^{(\pi_r)}(d) = \frac{1}{M} \sum_{i=1}^{M} \mathrm{SEP}_i^{(\pi_r)}(d). \tag{19}$$

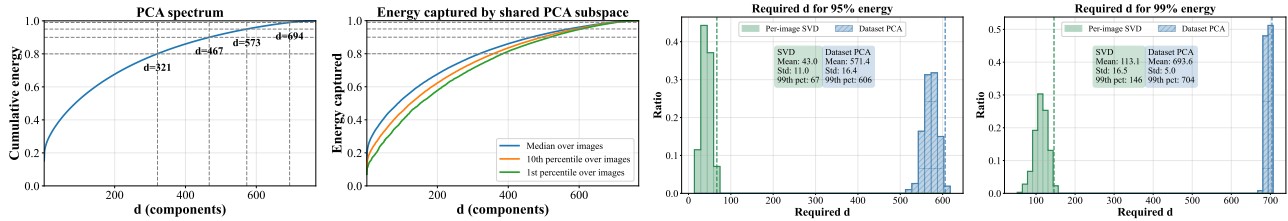

*Figure 9.* ViT-Base (DINOv2).

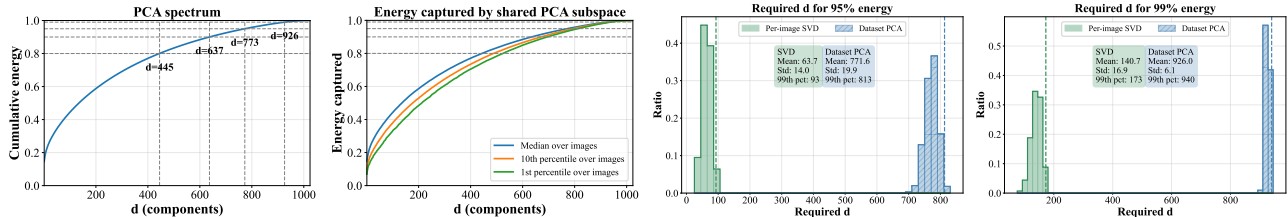

*Figure 10.* ViT-Large (DINOv2).

As summary statistics, we report the normalized curve distance

$$\Delta_{\text{curve}}^{(\pi_r)} = \frac{1}{100D} \sum_{d=1}^{D} \left| \text{S\={E}P}^{(\pi_r)}(d) - \text{S\={E}P}(d) \right| \tag{20}$$

and the bandwidth shift

$$\Delta b_\alpha^{(\pi_r)} = \frac{1}{M} \sum_{i=1}^{M} b_{\alpha,i}^{(\pi_r)} - \frac{1}{M} \sum_{i=1}^{M} b_{\alpha,i}. \tag{21}$$

For uncertainty, we use a paired bootstrap over images with 2,000 resamples.

The qualitative SEP law is robust to random global channel reorderings. Across the 14 transformer-family models considered in this paper and appendix, the permutation-mean SEP curves remain close to diagonal, with mean curve $L_1$ distances ranging from 0.0012 to 0.0059. The largest absolute bandwidth shifts are $|\Delta b_{80}| = 0.0073$ and $|\Delta b_{90}| = 0.0036$, both for ViT-Base (MAE). For the main-paper architectures, the shifts remain similarly small: CaiT-S24 $(-0.0037, -0.0007)$, DeiT-Small $(+0.0062, -0.0002)$, Swin-Small $(+0.0011, +0.0002)$, ViT-Large $(-0.0002, +0.0015)$, ViT-Huge $(-0.0032, -0.0023)$, and ViT-Tiny $(+0.0070, +0.0028)$ for $(\Delta b_{80}, \Delta b_{90})$. Therefore, while SEP is not permutation-invariant as a mathematical object, the empirical broad-band energy-spread conclusion is not an artifact of one particular native channel ordering.

## D. Cost and Parameter-Matched Control

The proposed endpoint fixes add different amounts of capacity. Table 10 reports the updated parameter counts and FLOPs relative to DeiT-Tiny. Lift is a lightweight endpoint projector, adding 266,112 parameters ($+4.65\%$) and increasing FLOPs from 2.507G to 2.536G ($+1.17\%$). WideLast widens only the final block, adding 1,521,600 parameters ($+26.61\%$) and increasing FLOPs to 3.089G ($+23.2\%$).

More importantly, the gains are not explained by parameter count alone. We train a parameter-matched control, **Uniform219dim**, which widens every layer of DeiT-Tiny to dimension 219 and distributes the extra capacity across the whole network rather than concentrating it at the endpoint. Although Uniform219dim has a slightly larger parameter budget than WideLast (7,372,759 vs. 7,239,016 parameters), it underperforms WideLast both as a standalone model ($75.15\%$ vs. $75.54\%$) and under SpectralKD-only distillation ($75.61\%$ vs. $76.53\%$). This indicates that *where* capacity is placed matters: resolving the endpoint width mismatch is more effective than distributing similar capacity uniformly through the network.

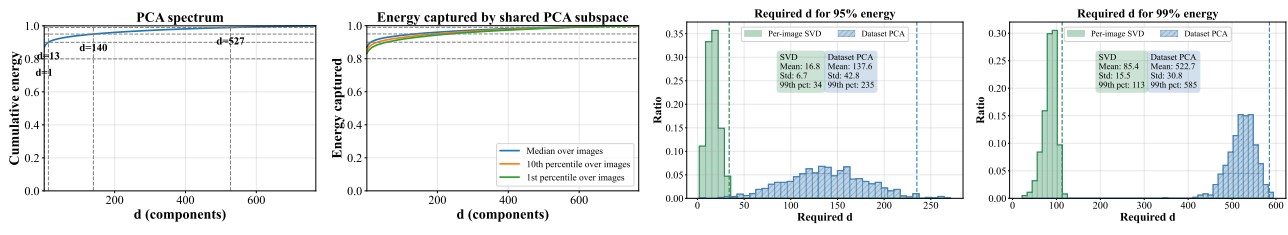

*Figure 11.* ViT-Base (MAE).

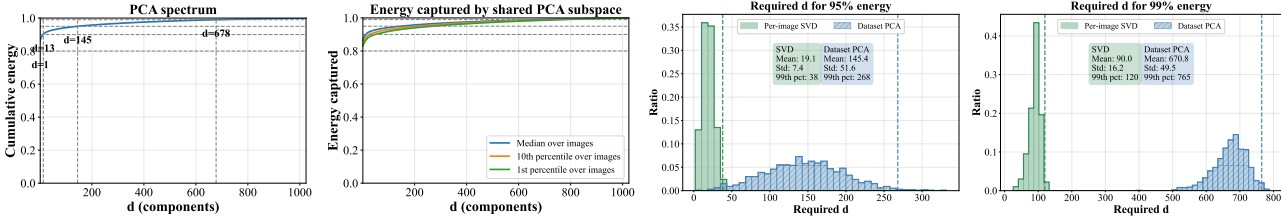

*Figure 12.* ViT-Large (MAE).

# E. A Brief Discussion about CNN

We extend the same endpoint diagnostics to standard CNNs. For a CNN final-stage feature map, we treat spatial locations as the token axis and channels as the channel axis, yielding a matrix in $\mathbb{R}^{HW \times C}$ for the SVD/PCA/SEP analyses. This makes the comparison directly analogous to the ViT endpoint analysis in the main text.

Figures 18–21 show that ResNet-18/34/50/101 exhibit the same qualitative representation geometry: individual images are low-rank under an input-specific SVD, whereas a shared dataset-level PCA subspace must remain high-dimensional to preserve most images. Figure 22 further shows near-diagonal SEP curves for all four CNNs, with approximately 80%/90% energy requiring roughly 0.79/0.90 of the channel bandwidth. Thus the endpoint signature is not unique to ViTs.

The difference is mainly architectural. Many common CNN distillation pairs are already aligned at the endpoint: ResNet-18 and ResNet-34 both use 512-dimensional final features, while ResNet-50 and ResNet-101 use 2048-dimensional final features. This helps explain why final-layer feature matching can appear more reliable in standard CNN settings. In the controlled FitNet experiment in Table 11, the matched-width ResNet-34 → ResNet-18 setup reaches 71.04%, whereas the mismatched ResNet-50 → ResNet-18 setup reaches 70.81%. The margin is small, so we do not claim this alone as a strong accuracy result; rather, it is consistent with the broader endpoint-mismatch interpretation and with prior observations that mismatched CNN feature alignment can be brittle.

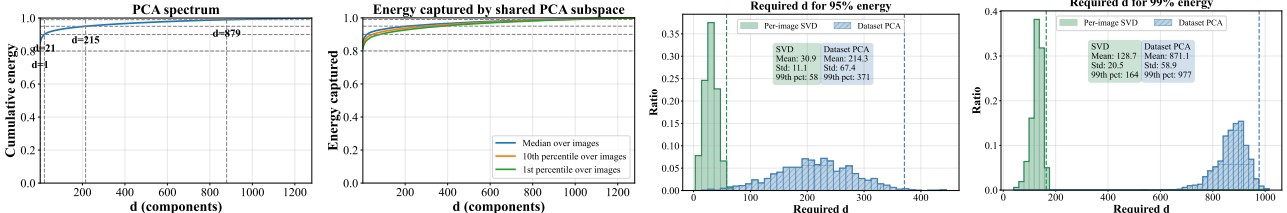

*Figure 13.* ViT-Huge (MAE).

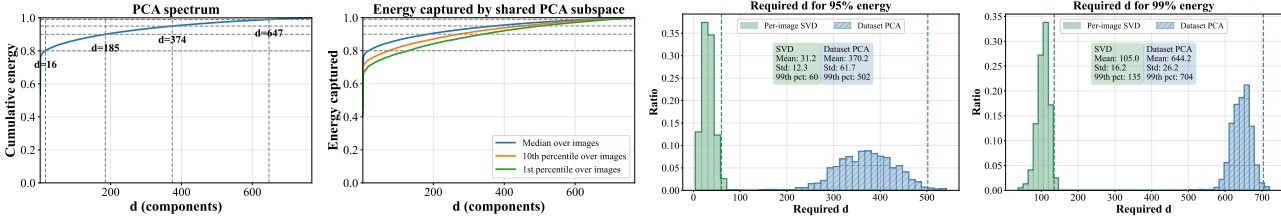

*Figure 14.* ViT-Base (CLIP).

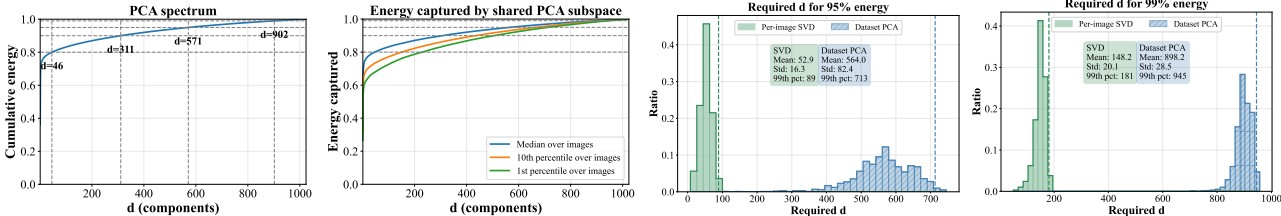

*Figure 15.* ViT-Large (CLIP).

*Table 10.* Cost and parameter-matched control on ImageNet-1K. Parameter/FLOP overheads are relative to DeiT-Tiny; SpectralKD-only uses the CaiT-S24 teacher.

| Variant | Placement | Params | FLOPs | w/o KD | SpecKD |
|---|---|---|---|---|---|
| Baseline | – | 5,717,416 | 2.507G | 74.86 | 75.07 |
| Lift | endpoint | 5,983,528 (+4.65%) | 2.536G (+1.17%) | 75.41 | 76.40 |
| WideLast | final block | 7,239,016 (+26.61%) | 3.089G (+23.2%) | 75.54 | 76.53 |
| Uniform219dim | all layers | 7,372,759 (+28.95%) | 3.195G (+27.4%) | 75.15 | 75.61 |

*Table 11.* CNN final-layer feature distillation on ImageNet-1K. The matched-width CNN teacher improves more than the mismatched-width teacher, but the margin is small.

| Student | Teacher | Feature KD | Top-1 (%) | Δ |
|---|---|---|---|---|
| ResNet-18 | None | None | 70.66 | – |
| ResNet-18 | ResNet-34 (512→512) | FitNet | 71.04 | +0.38 |
| ResNet-18 | ResNet-50 (2048→512) | FitNet | 70.81 | +0.15 |

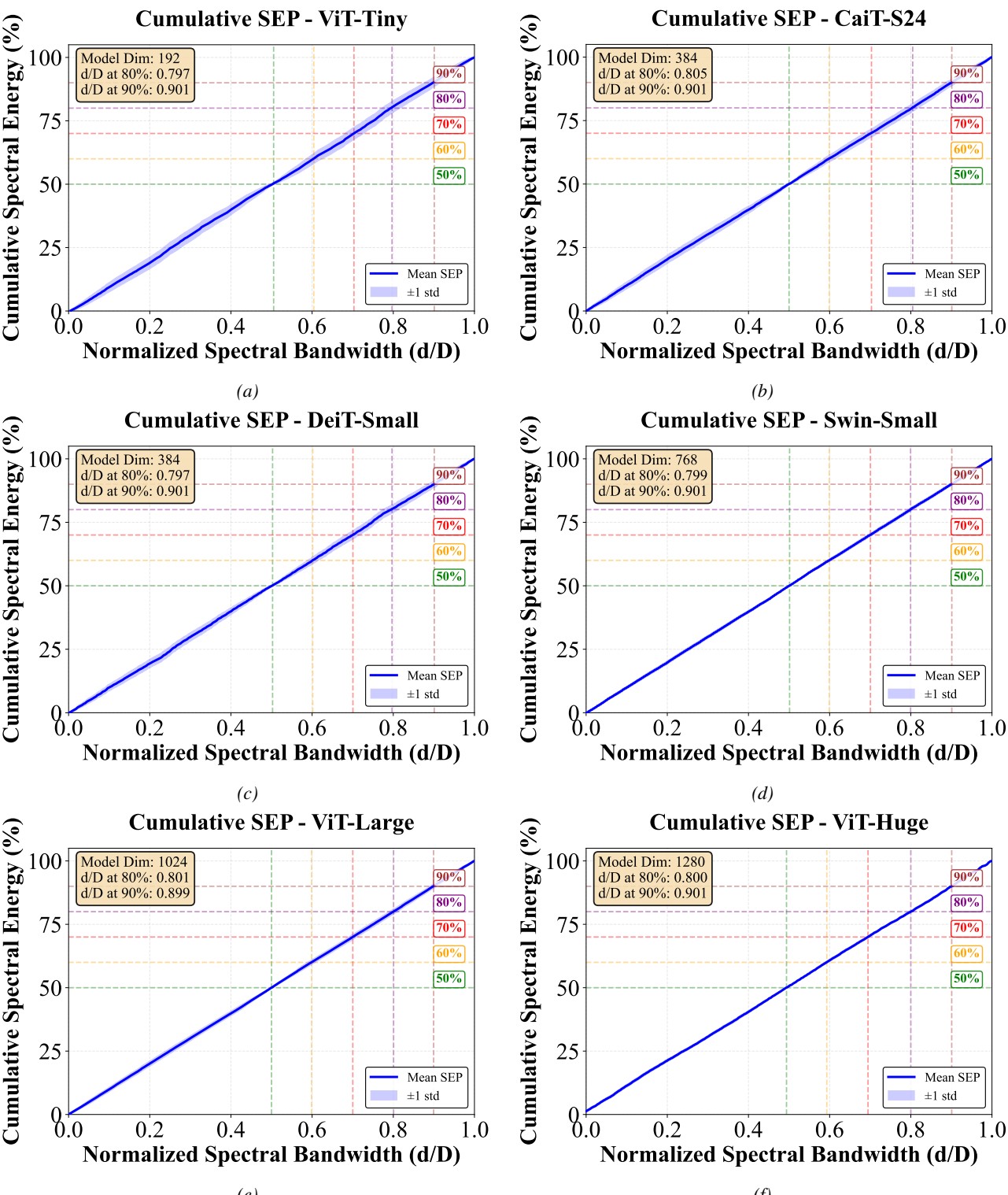

*Figure 16.* Spectral Energy Pattern (SEP) with mean and standard deviation across architectures. Each panel shows, for one model (ViT-Tiny, CaiT-S24, DeiT-Small, Swin-Small, ViT-Large, ViT-Huge (MAE)), the cumulative spectral energy as a function of normalized bandwidth $d/D$ together with a ±1 std band across images and reference lines at 50%, 60%, 70%, 80%, and 90% energy. All models exhibit almost perfectly diagonal SEP curves with very small variance, confirming that tokens consistently spread their energy across most available channel modes despite strong sample-wise compressibility of the token matrix under SVD.

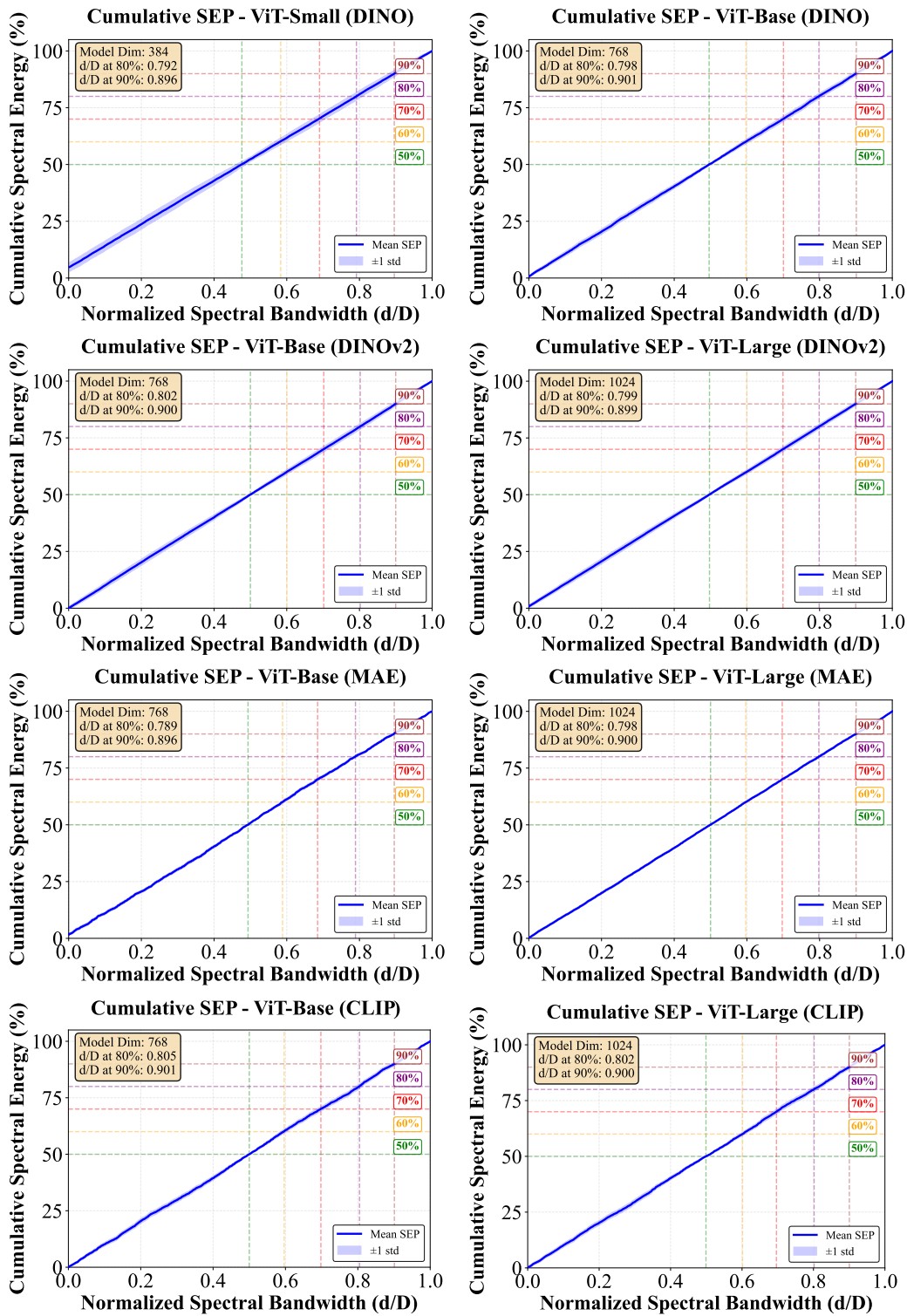

*Figure 17.* Additional per-model SEP curves for pretrained/self-supervised ViTs and CLIP visual encoders. Panels (left-to-right, top-to-bottom): ViT-Small (DINO), ViT-Base (DINO), ViT-Base (DINOv2), ViT-Large (DINOv2), ViT-Base (MAE), ViT-Large (MAE), ViT-Base (CLIP), and ViT-Large (CLIP).

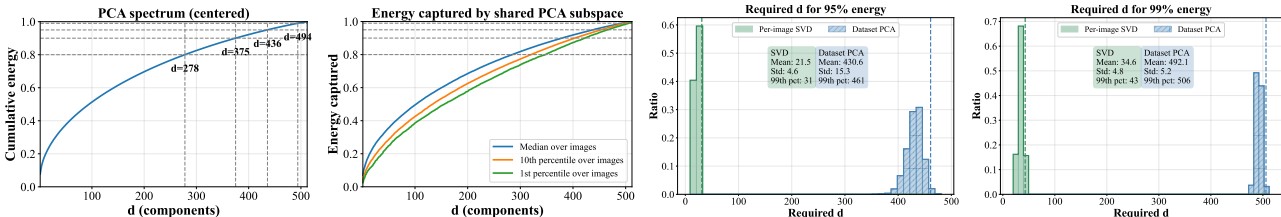

*Figure 18.* ResNet-18 SVD/PCA diagnostics. Spatial positions are treated as tokens and channels as the feature dimension.

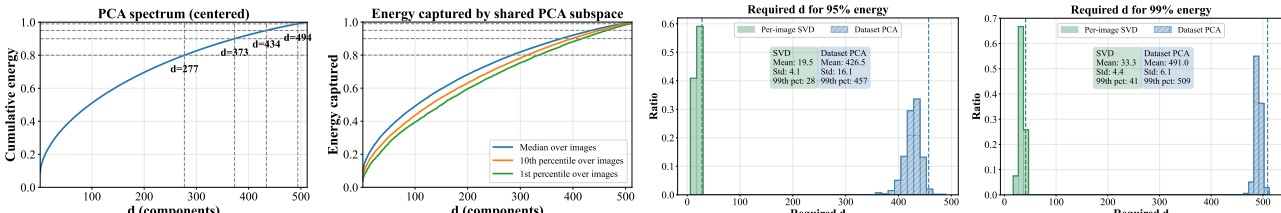

*Figure 19.* ResNet-34 SVD/PCA diagnostics.

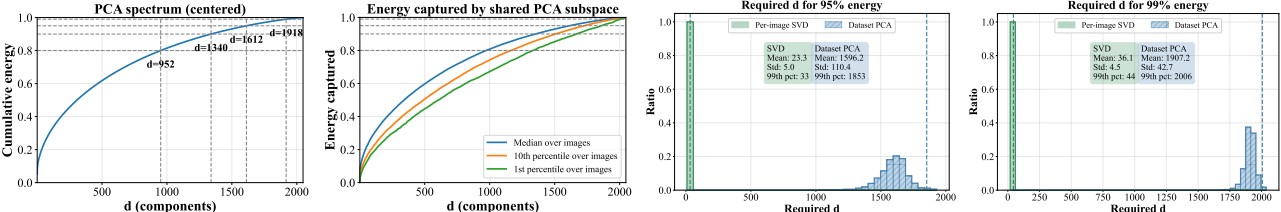

*Figure 20.* ResNet-50 SVD/PCA diagnostics.

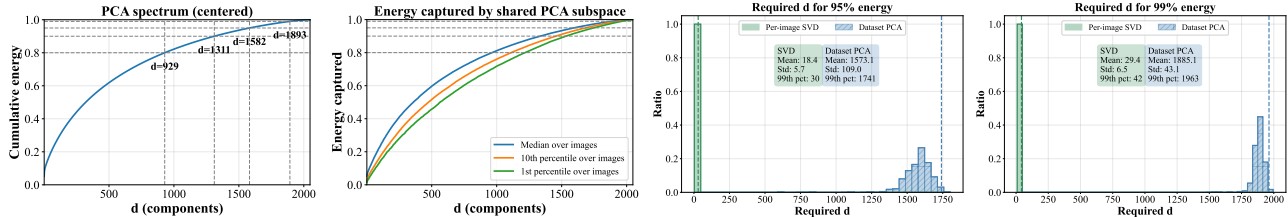

*Figure 21.* ResNet-101 SVD/PCA diagnostics.

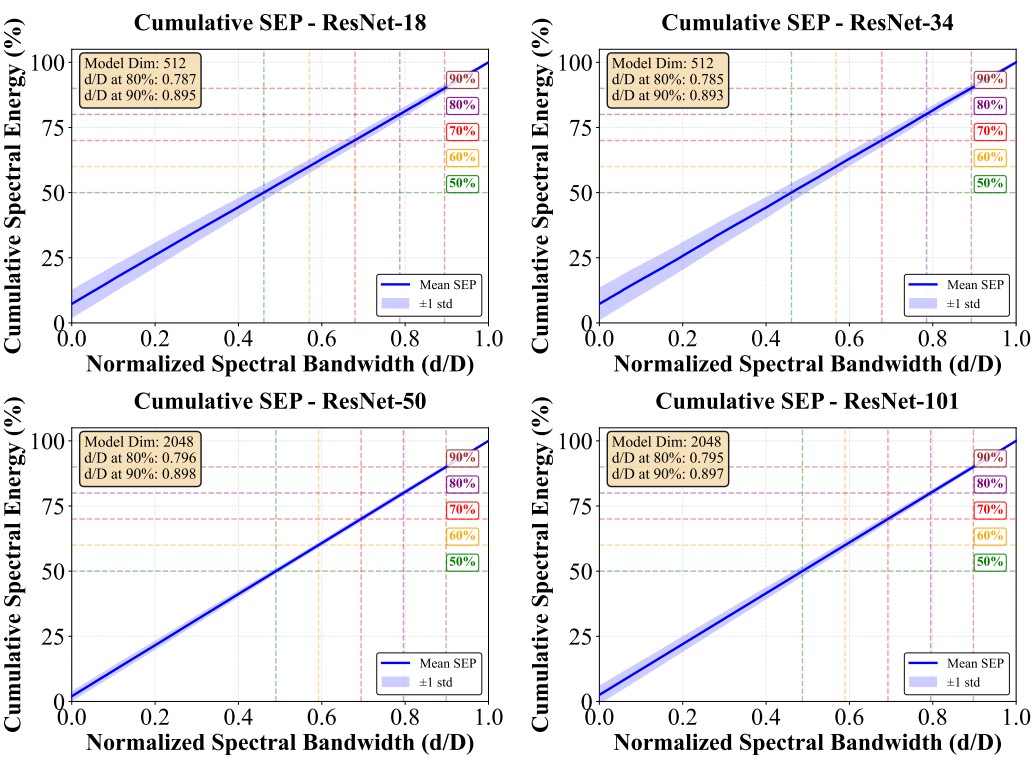

*Figure 22.* Additional per-model SEP curves for ResNet-18, ResNet-34, ResNet-50, and ResNet-101. Each panel plots cumulative spectral energy against normalized bandwidth $d/D$ with mean and standard-deviation bands over images.

