# OpenReview forum: "From Per-Image Low-Rank to Encoding Mismatch: Rethinking Feature Distillation in Vision Transformers"
_ICML.cc/2026/Conference — ICML 2026 regular_

### Official Review · Reviewer_UjkF · 2026-03-06

**Soundness:** 2
**Presentation:** 3
**Significance:** 3
**Originality:** 2
**Overall Recommendation:** 4
**Confidence:** 4

**Summary:**

This paper investigates why feature-based knowledge distillation (KD) often underperforms when compressing a wide Vision Transformer (ViT) teacher into a narrower student. The authors attribute this to an encoding mismatch at the final representation layer. To support this claim, they analyze teacher token representations from two complementary perspectives. First, per-image singular value decomposition (SVD) suggests that the token-feature matrix for an individual image is highly compressible. Second, dataset-level principal component analysis (PCA) shows that a single shared low-dimensional subspace is insufficient to represent most samples without retaining nearly the full feature dimension. The authors interpret this discrepancy as evidence that the effective representation subspace varies substantially across inputs.

The paper further introduces a token-level Spectral Energy Pattern (SEP) analysis and claims that token representations spread energy broadly across channels, implying a bandwidth mismatch in addition to the subspace mismatch. Based on these observations, the authors conclude that a narrow student equipped with a fixed linear projection head has difficulty matching the teacher’s representation space.

To address this issue, the authors propose two simple modifications at the student endpoint: Lift, which adds a lightweight projector that expands the student representation to the teacher width, and WideLast, which widens only the final transformer block of the student. Experiments on ImageNet-1K, primarily using a DeiT-Tiny student and a CaiT-S24 teacher, show that these modifications substantially improve feature distillation performance over the baseline setup, while also providing smaller gains even without distillation.

**Compliance With Llm Reviewing Policy:**

Affirmed.

**Final Justification:**

The paper presents an interesting analysis of feature-based knowledge distillation in Vision Transformers, highlighting a discrepancy between per-sample low-rank structure and dataset-level high dimensionality, and proposing simple architectural modifications (Lift and WideLast) to address this issue. The empirical results are strong, and the proposed modifications are simple and practically useful.

In my original review, I raised concerns regarding the robustness of the SEP analysis, the role of increased model capacity in the proposed improvements, and the limited experimental scope supporting the generality of the claims. The authors have addressed these concerns in the rebuttal through additional experiments and analyses, including robustness checks, capacity-controlled comparisons, and expanded empirical evidence. These additions substantially strengthen the technical support for the paper’s claims.

While the contribution remains somewhat incremental in terms of architectural novelty and some limitations remain in scope and generality, I believe the paper is now technically sound and provides useful insights and practical improvements for distillation in transformer models. Based on the rebuttal, I am updating my recommendation to Weak Accept.

**Key Questions For Authors:**

1. **Robustness of the SEP analysis.**
   The SEP analysis applies a Fourier transform across the channel dimension, even though channel ordering in transformer representations is not inherently meaningful. How robust are the SEP results to random permutations of the channel dimension?

2. **Capacity vs. alignment in WideLast.**
WideLast increases the width of the final transformer block, which also increases the number of parameters and computational cost. To what extent do the improvements come from better representation alignment rather than simply increased model capacity? Could the authors compare WideLast against parameter- or FLOP-matched alternatives (e.g., slightly widening the student uniformly or adding an additional narrow block)?

3. **Layer-wise generality of the mismatch.**
   The analysis focuses primarily on the final-layer representations. Have the authors examined whether similar subspace mismatch phenomena occur at intermediate layers?

4. **Generality across architectures and compression ratios.**
   How does the proposed encoding mismatch explanation generalize across different teacher–student width ratios and across transformer architectures within the same family?

5. **Relation to prior projection-based distillation methods.**
   Could the authors clarify how **Lift** differs from or improves upon standard projection heads used in prior feature-based distillation methods?

**Limitations:**

The paper includes a short impact statement, but the discussion of limitations and potential societal impacts is minimal. The authors state that the goal of the work is to advance machine learning and that no specific societal consequences need to be highlighted. While the work appears primarily methodological, the paper would benefit from a brief discussion of the limitations of the current study and the scope of the conclusions.

**Strengths And Weaknesses:**

### Strengths

1. **Clear empirical observation and motivation.**
The paper highlights an interesting discrepancy between per-sample low-rank structure (observed via SVD) and dataset-level high dimensionality (observed via PCA). This provides an intuitive way to frame representation variability in Vision Transformer features and motivates the proposed investigation.

2. **Simple and practical remedies.**
The proposed modifications, **Lift** and **WideLast**, are straightforward architectural adjustments that are easy to implement and integrate into existing distillation pipelines.

3. **Strong empirical improvements.**
The reported gains on ImageNet-1K are substantial for the studied setting (e.g., several percentage points improvement for the DeiT-Tiny student distilled from CaiT-S24), suggesting that the proposed modifications can provide practical benefits.

4. **Insightful representation analysis.**
The SVD and PCA analyses provide an intuitive perspective on why a fixed linear projection between student and teacher representations may be insufficient when compressing wide transformer models.

5. **Clear overall narrative.**
The paper presents a coherent progression from empirical observation, to representation analysis, to architectural modifications that attempt to address the identified issue.

---

### Weaknesses

1. **Causal interpretation is not fully established.**
The paper argues that the observed “encoding mismatch” explains the failure of feature-based distillation in wide-to-narrow compression. However, the evidence primarily demonstrates correlations rather than isolating this mechanism as the main cause. Other factors, such as optimization challenges or architectural differences between teacher and student, could potentially contribute to the observed behavior.

2. **Limited justification for the spectral energy analysis.**
The Spectral Energy Pattern (SEP) analysis applies a Fourier transform across the channel dimension. Since the ordering of channels in transformer representations does not necessarily correspond to a meaningful spatial or frequency structure, the interpretation of spectral bandwidth along this dimension is not fully justified.

3. **Architectural modifications increase model capacity.**
The proposed fixes alter the student architecture. In particular, **WideLast** widens the final transformer block, increasing parameters and computational cost, and **Lift** retains an additional projection layer at inference. It is therefore unclear how much of the improvement arises from better representation alignment versus increased model capacity.

4. **Limited comparison with strong distillation baselines.**
The experiments mainly compare against basic feature matching and logit distillation. The literature on transformer distillation includes several stronger baselines (e.g., attention-based or relational distillation methods), and additional comparisons would help better position the proposed approach.

5. **Related work not included in the main paper.**
Most of the related work discussion is moved to the supplementary material. This makes it harder to assess novelty and positioning relative to prior work directly from the main paper.

6. **Limited diversity of experimental settings.**
Most experiments focus on a single teacher–student configuration (CaiT-S24 to DeiT-Tiny). Evaluating additional architectures or compression ratios would help demonstrate whether the proposed explanation generalizes more broadly.

---

> ### Author Rebuttal · Authors · 2026-03-30
>
> ## Response to Reviewer UjkF
>
> Thank you for the constructive review. We agree that the initial submission did not fully establish several important points. We address them point-by-point below and will revise the manuscript accordingly.
>
> **Q1 & W2: SEP robustness to channel permutation.**
> > We tested SEP under 100 global random channel permutations on the same 1,000-image feature sets across 14 transformer-family models. The result is very stable: the mean-curve L1 distance is only 0.0012–0.0059; the largest shift is +0.0073 for b80 and +0.0036 for b90; for CaiT-S24 specifically, $b_{80}$ changes from 0.8047 to 0.8010 and $b_{90}$ from 0.9021 to 0.9014. Thus the near-diagonal SEP law persists under arbitrary global reorderings (see https://anonymous.4open.science/r/ICML26383/sep_permutation.png).
>
> **Q2 & W3: capacity increase vs. better alignment.**
> > We added `Uniform219dim`, a parameter/FLOP-matched control that distributes approximately the same extra budget as WideLast across all layers. It is weaker than `WideLast` both without distillation (75.15% vs. 75.54%) and with distillation (75.61% vs. 76.53%). This suggests that the gain is not explained by raw capacity alone; where the extra capacity is placed matters.
>
> > Consistently, even with only a marginal increase in model size (+4.65% parameters) and compute cost (+1.17% FLOPs), `Lift` already enables simple feature-based knowledge distillation (KD) to deliver consistent performance gains. We will clarify this in the paper: Lift mainly addresses endpoint bandwidth, while WideLast also provides an input-dependent final mapping that can better adapt to rotated subspaces.
>
> **Q3 & W1: layer-wise generality and causal interpretation.**
> > We added layer-wise dataset-level PCA for both CNNs and ViTs (take ResNet-101 and DeiT-Small as examples, see https://anonymous.4open.science/r/ICML26383/resnet101_pca_layers.png and https://anonymous.4open.science/r/ICML26383/deit_small_pca_layers.png).
>
> > In **both** families, the required shared dimension increases with depth, especially at high energy thresholds. So the mismatch is not a final-layer artifact; rather, it becomes strongest near the endpoint. We will also soften the causal claim accordingly: encoding mismatch is not the only possible reason feature KD can fail, but it is a directly supported mechanism that explains why naive endpoint feature matching often fails in wide-to-narrow compression.
>
> **Q4–Q5 & W6: generality across architectures and relation to prior projection-based methods.**
> > We added **both** same-family and CNN evidence.
> >
> > 1. Using the unchanged DeiT baseline, direct feature-map distillation from DeiT-Small and DeiT3-Small-21K yields 74.52% and 74.10%, versus the 74.86% baseline, showing that **negative transfer** can occur even within the DeiT family.
> >
> > 2. For CNNs, the final layers of ResNet-18/34/50/101 show the **same** qualitative pattern: per-image low rank, dataset high rank, broad SEP (take ResNet 101 as an example, see https://anonymous.4open.science/r/ICML26383/resnet101.png).
> >
> > 3. Besides, simple feature distillation of ResNet-34 → ResNet-18 (512 → 512, 71.04% Top 1 Acc on ImageNet) outperforms ResNet-50 → ResNet-18 (2048 → 512, 70.81% Top 1 Acc on ImageNet), consistent with stronger endpoint mismatch making direct feature alignment harder.
>
> > We will also clarify that `Lift` differs from standard training-only projection heads because its projector is **retained at inference**, while `WideLast` internalizes the expansion in the last block and makes it input-dependent.
>
> **W4 & W5: stronger baseline positioning and related work.**
> > We agree that stronger baselines and clearer positioning are important. We therefore aligned the experimental setup for ManifoldKD [1] and MaskedKD [2] to the same teacher/student pair (CaiT-S24 -> DeiT-Tiny). Our simple endpoint modifications reach 77.5% and 78.2%, compared with 76.5% for ManifoldKD [1] and 75.9% for MaskedKD [2]. We will also move more related-work discussion into the main paper.
>
> > [1] Learning Efficient Vision Transformers via Fine-Grained Manifold Distillation (NeurIPS 2022)
> >
> > [2] The Role of Masking for Efficient Supervised Knowledge Distillation of Vision Transformers (ECCV 2024)
>
> Thank you again for the helpful suggestions. These additions substantially strengthen both the empirical support and the positioning of the paper.

---

> > ### Author Rebuttal · Reviewer_UjkF · 2026-04-01
> >
> > I thank the authors for their detailed rebuttal. Based on the additional clarifications and experiments provided, I am updating my recommendation to Weak Accept.

---

> > > ### Author Response · Authors · 2026-04-01
> > >
> > > We sincerely appreciate your careful evaluation of our rebuttal and your updated recommendation. Your insightful comments have significantly helped us improve the quality and clarity of our manuscript. Thank you again for your time and support!

---

### Official Review · Reviewer_wJ1M · 2026-03-11

**Soundness:** 2
**Presentation:** 3
**Significance:** 2
**Originality:** 2
**Overall Recommendation:** 2
**Confidence:** 3

**Summary:**

This paper revisits feature-map knowledge distillation in Vision Transformers and examines why narrow students struggle to match wide teachers at the representation level. The paper identifies a geometric discrepancy between per-image and dataset-level structure: although token-feature matrices are highly low-rank on a per-image basis (via SVD), a shared low-dimensional subspace across the dataset (via PCA) requires nearly full dimensionality to preserve most spectral energy, which is interpreted as subspace rotation across samples. To further characterize this phenomenon, the paper introduces a Spectral Energy Pattern (SEP) analysis, revealing near-uniform channel-wise spectral utilization and suggesting high bandwidth usage within token representations. Based on these observations, the paper attributes feature distillation failure to an endpoint encoding mismatch caused by limited student channel capacity, and proposes two simple remedies: adding a lifting projection layer after the student’s final representation or widening the last transformer block. Experiments on ImageNet-1K demonstrate consistent improvements in distillation performance under these modifications.

**Compliance With Llm Reviewing Policy:**

Affirmed.

**Key Questions For Authors:**

Can the authors provide parameter- and computation-matched comparisons (e.g., models with equal total parameter count and comparable FLOPs but different layer-wise allocations) to rigorously isolate the effect of encoding mismatch correction from improvements attributable solely to increased model size and computational complexity?

**Limitations:**

The paper does not explicitly discuss limitations of the proposed approach, particularly regarding increased inference cost and its impact on model efficiency.

**Strengths And Weaknesses:**

Strengths
(1) The paper formulates a clear geometric tension between per-image low-rank structure and dataset-level high-dimensional subspace requirements, offering a structured diagnosis of feature distillation failure in ViTs.
(2) The introduction of Spectral Energy Pattern (SEP) provides a complementary frequency-domain perspective that strengthens the analysis of channel utilization and representation bandwidth.
(3) The proposed Lift and WideLast modifications are architecturally minimal yet empirically effective, making the central hypothesis easy to test and reproduce.
(4) The empirical analysis spans multiple ViT training paradigms, suggesting that the observed representation phenomena are broadly applicable rather than model-specific.

Weaknesses
(1) The necessity of full spectral matching is not established.
The paper assumes that effective feature distillation requires approximating the teacher’s full spectral energy or bandwidth. However, distillation is a task-driven compression process, and preserving task-relevant information does not necessarily require reconstructing the full representation manifold. The paper does not demonstrate that high-frequency components or full energy retention are critical for downstream accuracy, leaving the core encoding mismatch claim insufficiently justified.
(2) Parameter and computational complexity are not properly controlled.
Both Lift and WideLast increase the number of parameters and computational cost at the endpoint, yet no parameter- or FLOPs-matched baselines are provided. Since increasing width alone improves standalone student performance, it remains unclear whether the reported gains stem from resolving encoding mismatch or simply from higher parameter count and computation.
(3) Limited methodological novelty.
The proposed method consists of adding a linear projection layer or widening the final transformer block, both of which are standard architectural modifications.  As such, the methodological contribution is limited and lacks clear novelty.

---

> ### Author Rebuttal · Authors · 2026-03-30
>
> ## Response to Reviewer wJ1M
>
> We sincerely thank you for recognizing the clarity of our geometric tension formulation, the value of our SEP analysis, and the empirical effectiveness of our proposed modifications.
>
> **W1: necessity of full spectral matching**
> > We agree that our claim should not be interpreted as requiring exact reconstruction of the teacher’s full spectrum. However, our focus is on uncovering the mechanics of **when and why feature-map distillation** fails in wide-to-narrow compression. A **fixed narrow endpoint** becomes unreliable when the teacher simultaneously exhibits (1) strong per-image low rank, (2) large dataset-level subspace rotation, and (3) high per-token channel occupancy. In this setting, naive feature-map matching gives negligible or negative transfer, whereas restoring endpoint width makes the **same simple feature losses** effective again.
>
> >To make this clearer, we added new experiments showing that direct feature-map distillation with a standard DeiT student remains detrimental with other teachers (DeiT-Small and DeiT3-Small 21K: 74.52% and 74.10% vs. 74.86% baseline), which is consistent with the encoding-mismatch diagnosis. In the revision, we will explicitly soften the wording and state that our evidence supports a **practical sufficiency and necessity claim for reliable feature alignment under width mismatch**, rather than a theorem that full spectral preservation is universally required.
>
> **W2 & Q: parameter and FLOPs controls**
>
> >You raised a crucial point regarding whether our gains stem from resolving the encoding mismatch or simply from adding parameters. We have added FLOPs/parameter analyses and a parameter- and computation-matched baseline to isolate this effect.
>
> >- **Lift is highly efficient:** The `Lift` method adds only 266K parameters (a 4.65% increase) and 0.029G FLOPs (a 1.17% increase) to the DeiT-Tiny baseline. Despite this negligible overhead, it circumvents the bandwidth bottleneck and improves top-1 accuracy to 77.53%.
> >
> >- **Computation-Matched Baseline for WideLast:** While `WideLast` adds more capacity (a 26.61% parameter and 23.2% FLOPs increase), we introduced a new baseline called `Uniform219dim` to control for this. We expanded every layer of DeiT-Tiny evenly to 219 dimensions, resulting in a slightly larger 29.0% parameter and 27.4% FLOPs increase.
> >
> >- **Results:** The `Uniform219dim` model only achieved 75.15% standalone and 75.61% after feature distillation. In contrast, `WideLast` achieved 75.54% standalone and 76.53% after distillation. This confirms that resolving encoding mismatch yields better results than naive, uniform network scaling of comparable computational cost.
>
> > While `WideLast` introduces computational overhead at the final block, `Lift` provides practitioners with a highly efficient alternative (less than a **1.2%** FLOP increase) to resolve the mismatch in resource-constrained environments.
>
> **W3: methodological novelty**
>
> >We deliberately designed `Lift` and `WideLast` to be **minimal**. The primary contribution of our paper is the theoretical and analytical framework to diagnose **where and why** feature alignment breaks down.
>
> >The simplicity of our fixes is intended to validate the accuracy of this diagnosis. By pinpointing the exact bottleneck, we demonstrate that basic architectural modifications can revitalize vanilla MSE feature distillation.
>
> >Furthermore, this theoretically grounded approach outperforms engineered, complex distillation modules. For instance, our simple methods reache 77.5%/78.2% accuracy, surpassing recent methods like Manifold KD (76.5%) and MaskedKD (75.9%). Our goal is to provide the community with an interpretable diagnostic lens applicable to broad feature alignment scenarios, rather than proposing another black-box distillation module.
>
> **Generality of the theoretical framework**
>
> >To further demonstrate the universality and methodological significance of our findings:
> >1. We have added comprehensive SVD, PCA, and SEP analyses of the ResNet family (ResNet-18 through ResNet-101) to the revision (take ResNet 101 as an example, see https://anonymous.4open.science/r/ICML26383/resnet101.png). We discovered that CNNs exhibit the exact **same** encoding pattern, proving this is a fundamental property of deep representations rather than ViT-specific.
> >
> >2. We also added layer-wise PCA analyses showing that effective rank increases with depth in **both** CNNs and ViTs, which helps unify several prior empirical findings on why earlier layers are easier to align than later ones (ResNet-101 and DeiT-Small as examples, see https://anonymous.4open.science/r/ICML26383/resnet101_pca_layers.png and https://anonymous.4open.science/r/ICML26383/deit_small_pca_layers.png).
>
> Thank you again for your useful guidance. They allow us to refine the SEP analysis, provide parameter/FLOPs controls, and better highlight the broad generality of our theoretical framework beyond the specific methods proposed.

---

> > ### Author Rebuttal · Reviewer_wJ1M · 2026-04-05
> >
> > Thanks for authors' response. I will keep my rating.

---

> > > ### Author Response · Authors · 2026-04-05
> > >
> > > Dear Reviewer wJ1M,
> > >
> > > Thank you for taking the time to review our rebuttal.
> > >
> > > We noticed the system status indicates "Partially resolved - I have follow-up questions for the authors". Since we did not find specific questions in your comment, we are writing to check if there are any remaining technical concerns we can clarify for you.
> > >
> > > Thank you again for your time and efforts.
> > >
> > > Kind regards,
> > >
> > > The Authors

---

### Official Review · Reviewer_wXtB · 2026-03-11

**Soundness:** 2
**Presentation:** 3
**Significance:** 3
**Originality:** 3
**Overall Recommendation:** 4
**Confidence:** 3

**Summary:**

The paper first investigates the structure of the final-layer token embeddings in ViTs. They show that for individual images, these embeddings approximately lie on a low-dimensional subspace. However, across a whole dataset, the embeddings nearly span the entire ambient space. From these insights, the authors then propose two approaches aimed to improve ViT distillation: 1) Lift, which adds a linear projector to the last student layer to project the student’s last-layer embedding space up to the teacher’s last-layer embedding space, and 2) WideLast, which replaces the student’s final transformer layer with one whose width matches that of the teacher’s. The authors include experiments and ablation studies evaluating their approaches.

**Compliance With Llm Reviewing Policy:**

Affirmed.

**Final Justification:**

The discussion with the authors answered my questions and concerns, and reinforced my initial positive evaluation. I find the work insightful, and I lean towards recommending acceptance.

**Key Questions For Authors:**

1. Could the authors clarify why “high-bandwidth [is] the quantity that matters for endpoint capacity” (see Weakness 1 above)? This would help support the need for the SEP analysis.

2. From Weakness 2, WideLast seems to add a significant number of parameters in the student model, which is all in the last layer. What happens if you “spread out” the increased number of parameters across all layers of the student, rather than concentrating them within the last transformer layer? Is the resulting accuracy comparable to WideLast, or does it degrade significantly? I think this would clarify if the effectiveness of WideLast is due to width alignment vs. added parameters. From the Lift results, I’m inclined to believe it’s due to the former, but I think including experiments on this would be more convincing.

**Limitations:**

yes

**Strengths And Weaknesses:**

Strengths:

1. The paper is well-written and presents its ideas clearly.
2. The empirical observations regarding the low / high-dimensionality of the last-layer token embeddings are insightful.
3. Experimental results regarding Lift and WideLast are convincing of their effectiveness, as they achieve noticeable accuracy improvements

Weaknesses:

1. It’s not clear what the SEP results add that can’t be concluded from the dataset-wide PCA insights. In particular, I’m not convinced of the claim “In particular, even if each image’s token matrix lies near a low-dimensional subspace (sample-wise low rank), individual tokens can still use rich, high-bandwidth encodings within that subspace, which is the quantity that matters for endpoint capacity.” My understanding is that Section 2.3 argues each individual token’s energy is equally distributed among $D’ \approx \frac{D}{2}$ unique frequency components, so all $D$ dimensions are needed. However, from Section 2.2, the tokens across the entire dataset span almost the entire ambient space $D$; it seems like this result alone should be sufficient in arguing that the full width $D$ is needed. Maybe I am missing or misunderstanding something.

2. Both Lift and WideLast add more parameters to the student model, and so it may not be super surprising that both approaches improve student accuracy. In particular, WideLast seems to add a non-negligible amount in the experiments: the entire last transformer block in DeiT-Tiny is replaced with one whose width matches CaiT-S24. From a quick calculation (which could be incorrect - please correct me if I’m wrong), WideLast increases the number of parameters in DeiT-Tiny’s last transformer layer from about 0.5M to about 1.5M. A 1M parameter increase is quite significant for a model that originally has 5.7M parameters. It’s not clear if the effectiveness of WideLast is due to the width alignment, or because of this increase in parameters. Since Lift also seems effective and only adds about a small number of parameters ($384 \times 192 \approx 73.7$K, which is about $1.2\%$ of $5.7$M), I think the experiments still support the authors’ main arguments. However, the authors should be more upfront about the added computation of their proposed approaches.

3. Table 6 seems inconsistent: 1) for Lift, only DeiT-Small is used for the teacher, but for WideLast, both DeiT-Small and DeiT3-Small 21K are teachers. It would make more sense to include DeiT3-Small 21K teacher results for Lift. 2) The baseline does not seem entirely fair; there are no distillation-only baselines (i.e. distillation without Lift or WideLast) using the two teachers.

---

> ### Author Rebuttal · Authors · 2026-03-30
>
> ## Response to Reviewer wXtB
> We thank the reviewer for the careful reading and constructive feedback.
>
> **W1 & Q1: why SEP is not redundant with dataset-level PCA.**
>
> > We agree that dataset-level PCA establishes an important aspect of the difficulty: a single shared subspace that works across all images must be wide. However, PCA and SEP address different questions. Dataset-level PCA measures the capacity required for a fixed shared subspace across inputs (i.e., global orientation and subspace rotation). In contrast, SEP measures the extent to which an individual token utilizes channel modes within the specific subspace its image occupies (i.e., per-token bandwidth utilization). Therefore, our claim is not that SEP replaces PCA, but rather that the endpoint mismatch consists of two coupled facets: (i) substantial subspace rotation across images, and (ii) dense per-token usage within each image’s active subspace. A narrow endpoint can fail due to either issue, and SEP is specifically designed to isolate the latter.
>
> > This distinction between PCA and SEP also explains the behavioral differences between `Lift` and `WideLast`: `Lift` primarily alleviates endpoint bandwidth mismatch through a fixed lifted interface, whereas `WideLast` additionally provides an input-dependent mapping that can better adapt to the rotated subspaces revealed by dataset-level PCA.
>
> > We also agree that our previous Fourier notation was not sufficiently clear. In the revision, we have simplified this explanation: SEP results are reported using the full FFT ordering and normalized bandwidth $d/D$. Our earlier discussion of $D'$ (or $\sim D/2$) was simply notational convenience for conjugate symmetry, rather than a separate compression claim. To address the concern that SEP may depend too strongly on channel ordering, we introduced a random global channel-permutation control. Across all tested models, the qualitative near-diagonal law persists: the permuted $b_{80}$ remains near 0.8 and $b_{90}$ near 0.9. This confirms that our broad-band SEP conclusion is robust (see https://anonymous.4open.science/r/ICML26383/sep_permutation.png).
>
> **W2 & Q2: width alignment versus added parameters.**
>
> > We agree that the parameter and computational overhead should be stated more explicitly. In our follow-up analysis, `Lift` adds 266,112 parameters (4.65% over DeiT-Tiny), while `WideLast` adds 1,521,600 parameters (26.61%). The corresponding FLOPs increases are 1.17% for Lift and 23.2% for WideLast.
>
> > More importantly, to test whether the gains of `WideLast` come merely from added parameter count, we trained a parameter-matched control that spreads the extra capacity across the whole network rather than concentrating it at the endpoint. Concretely, we widened every layer of DeiT-Tiny to dimension 219 (`Uniform219dim`), which is slightly higher than WideLast in parameter budget. This uniformly widened control underperforms `WideLast` both as a standalone model (**75.15% vs. 75.54%**) and under distillation (**75.61% vs. 76.53%**). This is a direct evidence that the improvement is not simply due to adding parameters, but to **where** the extra capacity is placed: matching the endpoint width is more effective than distributing similar capacity uniformly across the network.
>
> **W3: fairness of Table 6 and missing baselines.**
> > We appreciate this point and have conducted more experiments to Table 6.
> > 1. We agree that including the Lift + DeiT3-Small-21K setup would make the table more symmetric. Therefore, we added an experiment involving Lift + DeiT3-Small-21K.
> > 2. We added the missing **distillation-only baselines** on the plain DeiT-Tiny student, without Lift or WideLast. Using DeiT-Small and DeiT3-Small-21K as teachers, direct feature-map distillation gives **74.52%** and **74.10%**, respectively, versus the **74.86%** DeiT-Tiny baseline. These results show that, for this teacher-student width mismatch, naive feature-map alignment is not a strong baseline and can even produce **negative transfer**.
> >
> > Due to space constraints, only the newly added data from Table 6 is displayed here.
> |Model Configuration|Teacher|Feature KD|Top-1 (%)|$\Delta$ vs. Baseline|
> |:--|:--|:--|:--|:--|
> |DeiT-Tiny (Baseline)|None|None|74.86|-|
> ||DeiT-Small|SpectralKD Only|**74.52**|**-0.34**|
> ||DeiT3-Small 21k|SpectralKD Only|**74.10**|**-0.76**|
> |Lift|DeiT3-Small 21k|SpectralKD Only|**75.52**|**+0.66**|
>
> We thank you again for these helpful comments. They led us to clarify the role of SEP, make the computation/parameter tradeoff explicit, and add stronger controls showing that the benefit comes from resolving the endpoint mismatch rather than simply increasing model size.

---

> > ### Author Rebuttal · Reviewer_wXtB · 2026-04-01
> >
> > Thank you for the response - most of my concerns have been resolved. I have some follow-up comments / questions.
> >
> > 1. > `Lift` adds 266,112 parameters
> >
> > Based on the experimental description in the manuscript, my understanding is that the number of parameters in `Lift` should be $192 \times 384 = 73728$. Could you clarify this please?
> >
> > 2. > our claim is ... the endpoint mismatch consists of two coupled facets: (i) substantial subspace rotation across images, and (ii) dense per-token usage within each image’s active subspace. A narrow endpoint can fail due to either issue, and SEP is specifically designed to isolate the latter.
> >
> > Thank you for this clarification - it makes more sense to me that **both** issues can cause performance degradation in narrow output dimensions.  However, this also seems to slightly contradict the claim in the submission that "individual tokens can still use rich, high-bandwidth encodings within that subspace, **which is the quantity that matters for endpoint capacity.**" While I understand that this is the quantity that matters at the individual token level, the subspace rotations at the dataset level also play a role. To me, the bolded part in the quote makes it seem like the token-level endpoint mismatch is the **only** reason for performance degradation.
> >
> > 3. > `WideLast` additionally provides an input-dependent mapping that can better adapt to the rotated subspaces revealed by dataset-level PCA
> >
> > Do you think widening the whole last Transformer block is necessary to achieve this? I can see why `Lift` (a single linear layer) may not, but a simpler nonlinear transformation, e.g. a 2-layer MLP appended to the student with the output dimension matching the teacher's, may achieve something similar while being more parameter efficient than `WideLast`. Basically `Lift` but with an MLP instead of a linear projector. This is mostly for my own curiosity, and will not significantly impact my evaluation of the submission.

---

> > > ### Author Response · Authors · 2026-04-01
> > >
> > > We thank you very much for your careful and insightful follow-up comments.
> > >
> > > > **1. Clarification of the Lift parameter count.**
> > >
> > > Thank you for pointing this out. We agree that the manuscript should honestly distinguish between the parameter count of the **projector alone** and the **net parameter increase of the full Lift variant actually used in the experiments**. In our formulation, `Lift` consists of a token-wise linear projector $P \in \mathbb{R}^{D_S \times D_T}$ that is **retained at inference**, and the classifier head operates on the lifted $D_T$ -dimensional representation rather than the original $D_S$ -dimensional student endpoint. Therefore, if we counts only the newly inserted projector, the added parameters are indeed
> > > $$
> > > 192 \cdot 384 + 384 = 74{,}112.
> > > $$
> > >
> > > However, for the full `Lift` model used in our experiments, the classifier head also changes from $192 \times 1000 + 1000 = 193{,}000$ to $384 \times 1000 + 1000 = 385{,}000$. Hence the **net increase relative to the baseline student** is
> > > $$
> > > 74{,}112 + (385{,}000 - 193{,}000) = 266{,}112.
> > > $$
> > >
> > > We will revise the manuscript to state this distinction honestly, so that **74,112** refers to the projector alone, while **266,112** refers to the net parameter increase of the complete `Lift` variant.
> > >
> > > ---
> > >
> > > > **2. Clarification of endpoint capacity.**
> > >
> > > Thank you very much for your careful reading. We agree that our previous sentence was too absolute. Our intent was not to suggest that token-level bandwidth is the only reason for degradation. Rather, the manuscript’s actual claim is that SEP and PCA capture **complementary** aspects of the endpoint mismatch: SEP measures per-token channel utilization within an image’s active subspace, while dataset-level PCA captures how the relevant subspace itself varies across inputs.
> > >
> > > The paper also explicitly frames the failure mechanism as having two coupled facets, namely **orientation mismatch (rotation)** and **capacity mismatch (bandwidth)**. We therefore agree that the sentence “which is the quantity that matters for endpoint capacity” can be misread as exclusive. In the revision, we will change it to a more precise formulation such as:
> > >
> > > > “Individual tokens can still use rich, high-bandwidth encodings within that subspace, which is what matters for **token-level** endpoint capacity. In addition, dataset-level PCA captures the complementary issue that the relevant subspace rotates across inputs.”
> > >
> > > We hope this revised sentence better reflects our intended claim that **both** effects matter and that SEP is not a replacement for dataset-level PCA, but a complementary diagnostic.
> > >
> > > ---
> > >
> > > > **3. Whether `WideLast` is necessary, versus a simpler nonlinear adapter.**
> > >
> > > We appreciate this suggestion and agree that widening the whole last Transformer block is **not necessarily the only way** to obtain the benefit we attribute to `WideLast`. In our paper, `Lift` is a **fixed learned linear map** from the student endpoint to the teacher-width space, retained at inference as a shared interface across all inputs. Because the same linear map is applied to every image, it can increase endpoint bandwidth/capacity, but it cannot adapt its transformation to the input-dependent subspace rotations highlighted by our dataset-level PCA analysis. This is also consistent with the paper’s description of `Lift` as addressing the bandwidth side of the mismatch, while not resolving dataset-level subspace rotation.
> > >
> > > By contrast, `WideLast` replaces the endpoint with a teacher-width Transformer block, so it provides an **input-dependent** mapping that can realize different effective transformations for different images. That is the reason we described `WideLast` as better suited to accommodating the rotating subspaces revealed by PCA. At the same time, we do not think that a widened Transformer block is uniquely necessary.
> > >
> > > As your insightful suggestion, a lighter nonlinear adapter, such as a two-layer MLP appended to the student and producing teacher-width features, is a very reasonable and potentially more parameter-efficient alternative. We did not evaluate that variant in the current paper, so we prefer to present it as an interesting and promising future direction. This is also aligned with the paper’s conclusion that adaptive or input-dependent lifting is a promising next step.
> > >
> > > ---
> > >
> > > Thank you so much again for your valuable time and careful reading. It is a great encouragement to us.

---

### Official Review · Reviewer_ARVa · 2026-03-13

**Soundness:** 3
**Presentation:** 4
**Significance:** 3
**Originality:** 4
**Overall Recommendation:** 5
**Confidence:** 4

**Summary:**

This paper reveals the fact that the feature map of one image is low-rank, while the feature map of the whole dataset is not low-rank. They address this as the reason why knowledge distillation fails in the compression case. Based on this understanding, they design two remedies, Lift or WideLast. Experiment results show that these remedies can improve the performance.

**Compliance With Llm Reviewing Policy:**

Affirmed.

**Final Justification:**

My concerns are well addressed. Although the discussion on CNNs is not that convincing, since the performance difference is only marginal, I believe the authors would have a more reasonable discussion in the final version. I think overall this finding is valuable.

**Key Questions For Authors:**

see weakness.

**Limitations:**

yes

**Strengths And Weaknesses:**

Strengths:
- The paper is written in an interesting way, which attracts the reader to follow.
- The finding that each image is low rank but the dataset is not low rank is interesting and could explain many things.
- The two proposed remedies show improvements and prove that the understanding is correct.

Weakness:
- The paper mentioned that KD in CNN works well. Does it mean that for CNNs not only each image but also the dataset is low rank? In that case, how can CNNs do the classification? If a similar analysis of ViT in the paper can be provided on CNN as well and show the difference, this point will be stronger. Otherwise, this understanding may miss something.
- The understanding itself is interesting, if it is correct (which I believe). However, we cannot do many things on it since it basically shows that the compression is not realizable. Increase dimension in the last layer only provides a marginal improvement. If we increase the dimension in each intermediate layer then it is not compression. I am just trying to think about what we can do with this understanding. It's OK to ignore this point.
- The analysis is a bit limited, just on ViT for image classification tasks.

---

> ### Author Rebuttal · Authors · 2026-03-30
>
> ## Response to Reviewer ARVa
>
> We appreciate your thoughtful and encouraging review, and we are glad you found the per-image low-rank and dataset-level non-low-rank observations interesting.
>
> **W1: regarding CNNs**
>
> >We thank you for this insightful question. To address it, we extended our SVD/PCA/SEP analysis to ResNet-18/34/50/101. We observed the exact **same** qualitative endpoint signature as in ViTs: per-image low rank, dataset-level high rank, and high per-token/channel occupancy. For instance, the ResNet-101 results can be viewed here: https://anonymous.4open.science/r/ICML26383/resnet101.png. These findings suggest that the phenomenon identified in our paper is not specific to ViTs, but reflects a broader endpoint-encoding property inherent to CNNs as well.
>
> >These new CNN results help clarify why feature distillation has appeared more successful in standard CNN settings. A key practical difference is that many common CNN teacher-student pairs are **implicitly** aligned in endpoint dimensionality. For example, ResNet-18 and ResNet-34 **both use 512**-dimensional final features, whereas ResNet-50 and ResNet-101 use 2048-dimensional final features. Our additional distillation experiments support this: under simple final-layer feature alignment, ResNet-34 → ResNet-18 (512 → 512) achieves 71.04% Top-1 accuracy on ImageNet, whereas ResNet-50 → ResNet-18 (2048 → 512) **drops** to 70.81%. In other words, once the endpoint dimensionality is mismatched, CNN feature distillation also becomes less reliable.
>
> >This aligns with prior CNN literature, such as ReviewKD [1], naive single-layer feature matching under dimensional mismatch can be brittle or harmful (Table 1 and Table 2 in [1]). We will include these CNN analyses and experiments in the revised manuscript.
>
> **W2: regarding model compression**
>
> >We wish to clarify that our work does not imply model compression is infeasible. In pursuit of strong discrimination, trained models tend to utilize a large fraction of the available endpoint bandwidth. Once training finishes, this endpoint encoding pattern becomes relatively fixed. Consequently, **post hoc** feature compression becomes difficult for a substantially narrower student because it must approximate a representation that is both high-bandwidth and input-dependent in orientation.
>
> >However, this challenge pertains strictly to the feature space. The model's parameters may still contain substantial redundancy, leaving the model highly compressible at the parameter level. Our current results support this distinction: once the endpoint mismatch is mitigated by `Lift` or `WideLast`, a much smaller student can absorb the teacher's signal effectively. We will revise the manuscript to make the distinction between feature-space and parameter-space compressibility more explicit.
>
> **W2 (Continued): regarding practical applications**
>
> >Beyond knowledge distillation, our findings may offer practical utility for various tasks requiring feature-map alignment. For example, in multimodal models such as CLIP, if the visual encoder is narrower than the text encoder, selectively increasing endpoint capacity in the visual branch may improve cross-modal feature alignment while keeping the visual backbone compact.
>
> **W3: regarding scope and generality**
>
> >We agree that the original manuscript focused primarily on ViT image classification. To broaden the scope, we have introduced two extensions:
> >1. As detailed above, we conducted identical endpoint analyses on CNNs and observed the **same** qualitative SVD/PCA/SEP patterns.
> >
> >2. We performed layer-wise, dataset-level PCA analyses across both CNNs and ViTs. We found that the effective rank tends to increase gradually with depth in **both** model families (take ResNet-101 and DeiT-Small as examples, see https://anonymous.4open.science/r/ICML26383/resnet101_pca_layers.png and https://anonymous.4open.science/r/ICML26383/deit_small_pca_layers.png).
>
> >These results are significant because they unify several prior empirical observations in CNN and ViT distillation under a single analytical framework. For CNNs, ReviewKD [1] demonstrated that using shallower teacher representations to guide deeper student features is effective. For ViTs, works like ViTKD [2] and Distillation Dynamics [3] report that later-layer alignment can actually be detrimental. Our perspective is that these are not isolated heuristics, but phenomena that can be clearly understood through a common representation-level lens.
>
> >[1] Distilling Knowledge via Knowledge Review (CVPR 2021)
> >
> >[2] Vitkd: Feature-based knowledge distillation for vision transformers (CVPR 2024)
> >
> >[3] Distillation Dynamics: Towards Understanding Feature-Based Distillation in Vision Transformers (AAAI 2026)
>
> We sincerely appreciate your feedback, as it has substantially improved both the scope and the interpretation of our paper. We will revise the discussion to reflect this broader applicability.

---

> > ### Author Rebuttal · Reviewer_ARVa · 2026-04-06
> >
> > My concerns are well addressed. Although the discussion on CNNs is not that convincing, since the performance difference is only marginal, I believe the authors would have a more reasonable discussion in the final version. I think overall this finding is valuable.

---

> > > ### Author Response · Authors · 2026-04-06
> > >
> > > Dear Reviewer ARVa,
> > >
> > > **Thank you for recognizing our work and for your constructive feedback. Your suggestion to introduce CNNs for comparative analysis greatly inspired us and it broadened the generality of our theoretical findings.**
> > >
> > > Regarding your observation that the performance differences in CNNs are marginal, we will provide a more in-depth analysis of this in the discussion section of our final version. This phenomenon is closely related to the architectural characteristics of CNNs: unlike ViTs, which maintain a constant dimension across layers, CNNs typically expand their channel dimensions as the network deepens (e.g., the endpoint dimension of ResNet-50 reaches 2048), endowing them with inherently high endpoint capacity. As observed in our newly added controlled experiment (`Uniform219dim`), increasing the overall network dimension can naturally alleviate the "encoding mismatch" problem to some extent. However, this uniform expansion strategy is still not as effective as directly expanding the final layer (`WideLast`).
> > >
> > > We hope this cross-architecture comparative finding will provide valuable theoretical guidance for the design of more efficient ViT models in the future (e.g., introducing CNN-like dimension expansion mechanisms at the endpoint). In the revised manuscript, we will include more intuitive and detailed figures to rigorously present **the comparative analysis results of SVD, PCA, layer-wise PCA, and SEP across both CNN and ViT families**.
> > >
> > > Thank you once again for your efforts in helping us improve the quality of our paper!

---

### Decision · Program_Chairs · 2026-04-30

**Decision:**

Accept (regular)

**Comment:**

The reviewers found the paper to present an interesting and useful perspective on feature distillation in ViTs, supported by clear empirical observations and simple yet effective remedies. Most concerns were addressed in the rebuttal, including clarifications on SEP, stronger baselines, and parameter/FLOPs-matched comparisons, which strengthen the claim that the improvements are not solely due to increased capacity. One reviewer remained unconvinced regarding the strength of the causal interpretation and the limited methodological novelty. Overall, the paper provides a valuable diagnostic insight with solid empirical support, though the claims should be stated more cautiously.